# Growth-rate dependency of ribosome abundance and translation elongation rate in *Corynebacterium glutamicum* differs from that in *Escherichia coli*

Susana Matamouros [1] ✉, Thomas Gensch [2,4], Martin Cerff [1,4], Christian C. Sachs[1], Iman Abdollahzadeh[2], Johnny Hendriks[2], Lucas Horst[1], Niklas Tenhaef[1], Julia Tenhaef [1], Stephan Noack [1], Michaela Graf [3], Ralf Takors[3], Katharina Nöh [1] & Michael Bott [1] ✉

Bacterial growth rate (μ) depends on the protein synthesis capacity of the cell and thus on the number of active ribosomes and their translation elongation rate. The relationship between these fundamental growth parameters have only been described for few bacterial species, in particular *Escherichia coli*. Here, we analyse the growth-rate dependency of ribosome abundance and translation elongation rate for *Corynebacterium glutamicum*, a gram-positive model species differing from *E. coli* by a lower growth temperature optimum and a lower maximal growth rate. We show that, unlike in *E. coli*, there is little change in ribosome abundance for μ <0.4 h⁻¹ in *C. glutamicum* and the fraction of active ribosomes is kept above 70% while the translation elongation rate declines 5-fold. Mathematical modelling indicates that the decrease in the translation elongation rate can be explained by a depletion of translation precursors.

*Corynebacterium glutamicum* is a non-pathogenic, rod-shaped, gram-positive bacterium belonging to the class of *Actinomycetes* that is used for the production of amino acids in the million-ton-scale[1], but also for synthesis of various other metabolites and of proteins. Due to its outstanding importance in industrial biotechnology and non-pathogenic status, *C. glutamicum* is extensively used as model organism for the development of novel product platforms as well for studying selected topics that are relevant for pathogenic actinobacteria, such as *Corynebacterium diphtheriae* and *Mycobacterium tuberculosis*. As a result of its widespread use as microbial cell factory, *C. glutamicum*'s metabolism is one of the most studied in bacteria[2]. Fundamental physiological properties such as growth rate can have an important impact on the cell's metabolism and its productivity in bioprocesses[3]. Therefore, knowledge of the factors that determine and

limit the growth rate are of high interest from both a systems biological and a biotechnological view.

Our understanding of bacterial physiology includes a number of phenomenological growth laws[4–11] that describe how certain cellular parameters relate to the growth rate. One of these parameters is the ribosome abundance. As growth rate increases, usually in response to better nutrient quality, so does the ribosomal mass fraction[8,12]. Ribosomes are large multimeric RNA-protein complexes whose synthesis is tightly controlled to avoid loss of fitness due to resource misallocation[13–15]. A linear correlation between ribosome abundance and growth rate (Rb/μ) has been reported for several organisms, which highlights the importance of protein synthesis to cellular growth[10]. This proportionality relies on ribosomes translating at a constant rate[8]. Growth rate therefore depends on the number of active ribosomes

[1]Institute of Bio- and Geosciences, IBG-1: Biotechnology, Forschungszentrum Jülich, Jülich, Germany. [2]Institute of Biological Information Processing, IBI-1: Molecular and Cellular Physiology, Forschungszentrum Jülich, Jülich, Germany. [3]Institute of Biochemical Engineering, University of Stuttgart, Stuttgart, Germany. [4]These authors contributed equally: Thomas Gensch, Martin Cerff. ✉e-mail: s.matamouros@fz-juelich.de; m.bott@fz-juelich.de

(i.e., ribosomes engaged in peptide chain elongation) present in the cell. However, it is known that this linear correlation is only observed for moderate to fast growth rates ($\mu > 0.35\,h^{-1}$)[4], as a continuous linear decrease in the number of ribosomes as growth slows down would leave the cells with too few ribosomes to restart growth[5]. Most comprehensive studies on this subject have been performed on the fast-growing model organism *Escherichia coli*. In *E. coli* the translation elongation rate ($k$) decreases by about 50% from fast ($k = 16$–$17\,aa\,s^{-1}$, $\mu > 1\,h^{-1}$) to very low growth rates ($k = 9\,aa\,s^{-1}$, $\mu = 0.035\,h^{-1}$)[16–21], which shows that the translation elongation rate is not constant. The availability of aminoacyl-tRNAs as well as elongation factors and GTP may be reduced in certain conditions, such as during nutrient deprivation or by slow diffusion in the crowded cytoplasm[22–24], leading to a decrease in the elongation rate. Recently, Dai et al.[21] elegantly showed that *E. coli* is able to maintain faster-than-expected translation elongation rates during slow growth via a significant reduction of the pool of active ribosomes.

In this study, we systematically examined the ribosome distribution, abundance and activity for *C. glutamicum* across different growth rates. We find important differences in the fraction of active ribosomes during slow growth between *C. glutamicum* and *E. coli* that may reflect distinct evolutionary strategies for coping with periods of nutrient deprivation. The quantitative knowledge of these parameters for *C. glutamicum* extends the current *E. coli*-centred knowledge of these key cellular functions to a member of a phylogenetically distant group of bacteria, it enables the development of better growth models, and it can help to identify approaches for improving growth-coupled product formation with this important biotechnological host.

## Results

### Synthesis dynamics and spatial localization of ribosomes in *C. glutamicum* cells

To study the number and distribution of ribosomes in cells of *C. glutamicum* ATCC 13032 at different growth rates and growth stages (exponential and stationary), the wild-type strain (wt) was engineered to chromosomally encode for translational fusions between two ribosomal proteins, bL19 and uS2, and the fluorescent proteins EYFP and PAmCherry. These ribosomal proteins, which are located at the ribosome surface and incorporated at a late stage of ribosome assembly[25], were chosen because they have been successfully tagged with fluorescent proteins in *E. coli* and the resulting strains showed normal growth and no ribosome assembly defects[26,27]. In our engineered *C. glutamicum* strains, the protein fusions substitute the native bL19 and uS2 proteins. Synthesis of bL19-EYFP and uS2-PAmCherry in strain SM34 was

confirmed by western blot analysis of cell extracts (Supplementary Fig. 1). Strain SM34 grew similarly to the wt strain (Fig. 1a), suggesting that the two fusion proteins were successfully incorporated into the 50S and the 30S subunits, respectively, and did not interfere with ribosome functionality under the selected conditions. Online measurement of EYFP fluorescence in a BioLector cultivation system allowed us to follow ribosome synthesis dynamics during growth, namely that of the large ribosomal subunit. While no fluorescence was detected for the wt, for strain SM34 a fast increase in specific fluorescence (ratio of absolute fluorescence and cell density) was observed upon entry into the exponential phase followed by a decrease in the mid-exponential phase (Fig. 1a). This early burst in ribosome synthesis is very similar to the one described for *E. coli*[28,29].

In several bacterial species[26,30–33], ribosomes are preferentially found excluded from the highly condensed chromosomal DNA regions, the nucleoids. Therefore, to further verify that the observed fluorescent signal was ribosome-specific, wide-field fluorescence microscopy was performed to examine ribosome and nucleoid localization in exponentially growing and stationary *C. glutamicum* cells. The bL19-EYFP protein was found to be highly abundant in the cell (Fig. 1b), as expected for a ribosomal protein. However, it was not equally distributed throughout the cytoplasm, but formed areas of high and low densities, and the latter ones correlated with SYTOX orange-stained DNA-rich regions. In exponentially growing cells, a few DNA-rich regions can be distinguished whereas in stationary phase cells, the DNA is usually found in one highly condensed area (Fig. 1b). In both cases, ribosomes are found concentrated outside the DNA-rich areas. This spatial separation suggests that bL19-EYFP is incorporated into ribosomes that due to their large molecular size are less likely to diffuse into the condensed nucleoid areas.

### Ribosome quantification at the single-cell level

To quantify ribosome abundance in *C. glutamicum* cells, a single-molecule localization microscopy (SMLM) method was developed in which uS2-PAmCherry fluorescent molecules present in strain SM34 were quantified and taken as proxy for the amount of 30S ribosomal particles. For these experiments, we chose to work with PAmCherry as it allows for controlled fluorescence activation necessary for quantification[34]. Strain SM34 was grown either in batch culture using six different growth media or in chemostat culture with a fixed growth medium but varying dilution rates to yield a range of exponential growth rates ($\mu$) varying from $0.20\,h^{-1}$ to $0.49\,h^{-1}$. Immediately after sampling, cells were washed, fixed, and mounted on a home-built wide-field fluorescence microscope with single molecule sensitivity[35]. Cells

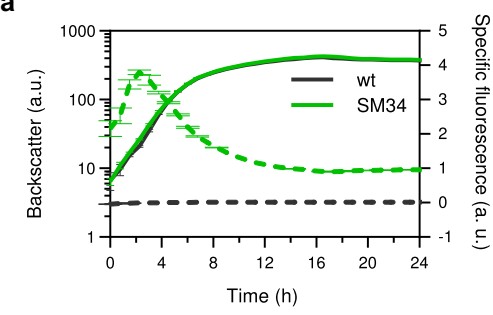

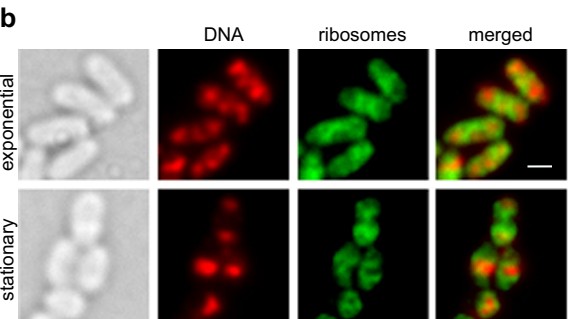

**Fig. 1 | *C. glutamicum* strains carrying fluorescently-tagged ribosomes show wt-like growth behaviour and ribosome-associated fluorescence. a** Growth curve (solid lines) and specific fluorescence (dashed lines) of the wt (black) and SM34 [(bL19-EYFP uS2-PAmCherry), green] strains from a BioLector cultivation in BHI + GLU medium. Shown is the mean and standard deviation ($n = 3$ biologically independent cultures). **b** DNA and ribosome distribution of exponentially growing and

stationary phase cells of *C. glutamicum* SM30 (bL19-EYFP) cultivated on BHI + GLU. Similar results were obtained for two independent experiments. The first pair of panels on the left are bright-field microscopy images. The three pairs of panels on the right are wide-field fluorescence microscopy images. The SYTOX Orange-stained nucleic acid is shown in red and the bL19-EYFP-labelled ribosomes are shown in green. Scale bar = 1 µm. Source data are provided as a Source data file.

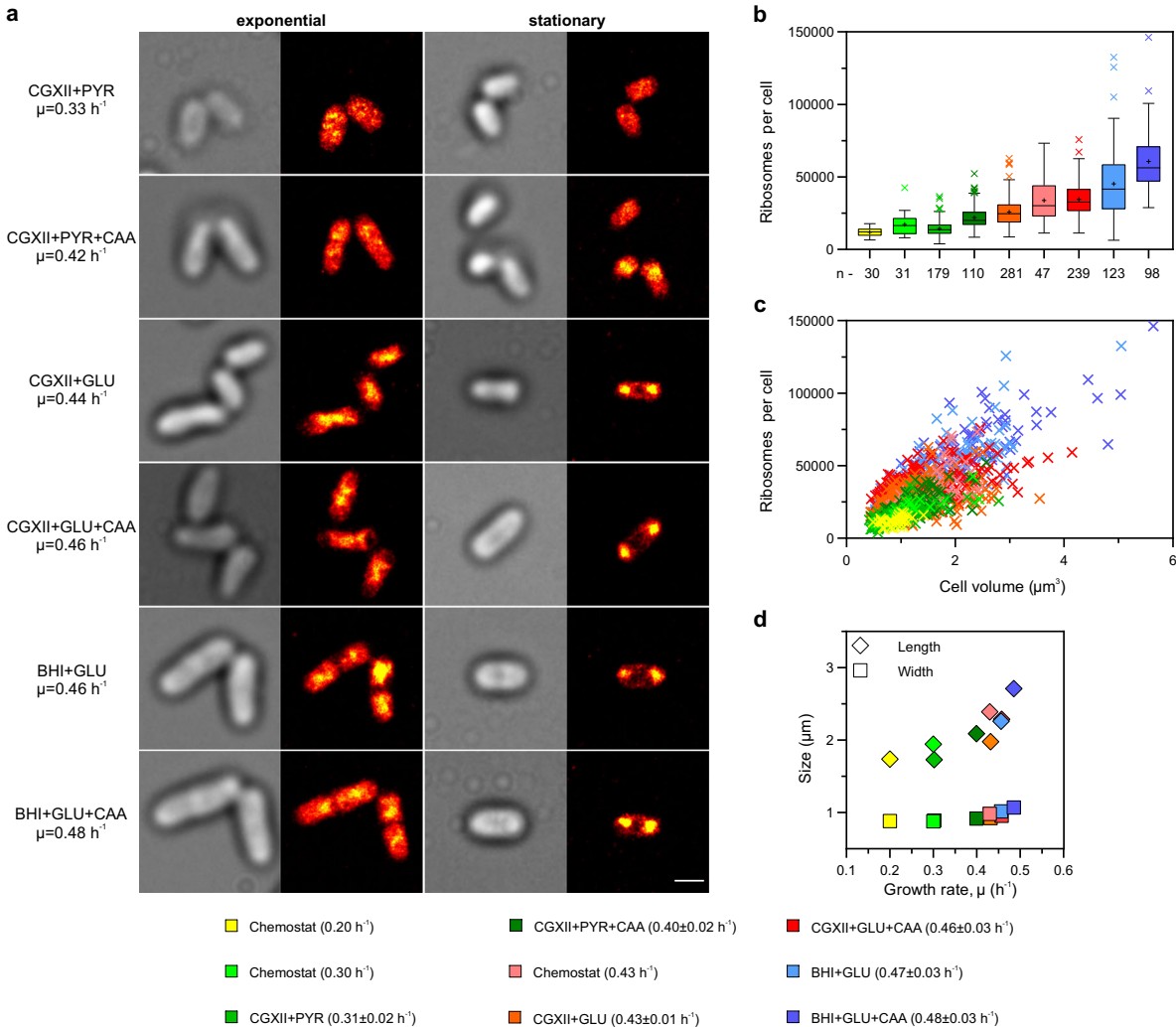

**Fig. 2 | Ribosome number per cell and cell volume increase with growth rate.**
**a** Examples of SMLM images taken from cells cultivated in different media and at different stages of growth. *C. glutamicum* SM34 was cultivated in different media enabling increasing exponential growth rates. Samples were taken either in mid-exponential or in stationary growth phase (24 h). Shown are the transmission image (grey) and the corresponding rendered SMLM image (colour) where each dot represents a PAmCherry localization. Higher density regions are shown in yellow whereas red and darker colours denote lower density regions. Scale bar = 1 μm. For each condition, the experiment was independently repeated at least twice and similar results were obtained. **b** PAmCherry counts (approximate number of ribosomes) per cell determined by SMLM for the different growth conditions shown in order of increasing growth rate. The data is depicted in a box and whiskers plot (Tukey method plotted in GraphPad Prism 9.1.2), where the bounds of the box correspond to the interquartile range (IQR) i.e. the difference between the 25th and the 75th percentiles. The whiskers stop at a data point value. The upper whisker equals the 75th percentile plus 1.5IQR and the lower whisker equals the 25th

percentile minus 1.5IQR. The middle line represents the median and the mean is shown by the + symbol. n indicates the number of cells examined over $n = 3$ biologically independent experiments for CGXII + PYR, CGXII + PYR + CAA, BHI + GLU and BHI + GLU + CAA or $n = 4$ biologically independent experiments for CGXII + GLU and CGXII + GLU + CAA. For the Chemostat controls only one ($n = 1$) sample was analysed. **c** PAmCherry counts (approximate number of ribosomes) determined by SMLM quantification in dependency of the cell volume are plotted for every individual cell analysed. Cell volume was calculated for every cell following the formula: $V = \pi * W^2(L - W/3)/4$, where W represent cell width and L cell length[78]. Growth conditions are colour-coded as in panel (**b**). For panels **b** and **c**, one outlier value for condition BHI + GLU is not shown in this representation (see Supplementary Fig. 3 for the full dataset). **d** The mean cell length (SEM is within 3–10% and approximately the symbol size) and width (SEM is within 1–2%) is shown for each growth condition (same colour code as in (**b**)) and plotted against the corresponding growth rate. Source data are provided as a Source data file.

were illuminated with low power 405 nm light for photoactivation and high power 561 nm light for single-molecule fluorescence initiation until all uS2-PAmCherry proteins were excited and detected.

Clearly distinct growth phase-dependent ribosome localization distribution was observed in the reconstructed SMLM images (Fig. 2a). In samples from the exponential growth phase, ribosomes were found throughout the bacterial cytoplasm with a few observable higher density areas. In stationary phase samples, especially in those where cells were cultivated in glucose-containing medium, ribosomes were preferentially found in high-density regions localized either at one or both cell poles (Fig. 2a) or, alternatively, mid-cell. Although the

nucleoid localization was not verified in these samples, these results suggest that as observed in the wide-field images, the ribosomes appear to be predominantly localized in areas outside the highly condensed nucleoid.

To determine the approximate number of ribosomes per cell, emitter molecules previously identified in reconstructed SMLM data via SNSMIL[36] were assigned to individual cells. To this end, the custom Super-resolution Emitter Counter software, *SurEmCo*, was developed (Supplementary Note 1). For each growth condition tested, the total number of uS2-PAmCherry fluorescent emitters per cell and the respective cell length and width were determined using *SurEmCo*

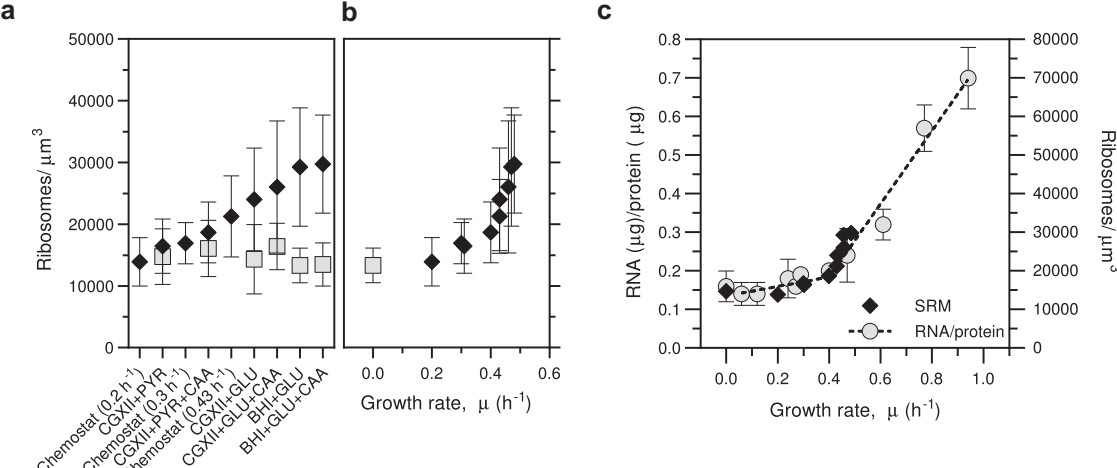

**Fig. 3 | *C. glutamicum* shows a nonlinear Rb/μ correlation. a** Ribosome density (PAmCherry counts per μm³ cell volume) as determined by SMLM quantification for the indicated nine growth conditions. Numbers obtained from exponentially growing cells are represented by black diamonds, those from stationary cells are shown as grey squares. Shown is the mean and standard deviation for each condition. The number of cells analysed and biologically independent repeats for the exponential growth condition is the same as in Fig. 2b. For the stationary growth samples, 113, 158, 264, 266, 239 and 137 cells were examined for conditions CGXII + PYR, CGXII + PYR + CAA, CGXII + GLU, CGXII + GLU + CAA, BHI + GLU and BHI + GLU + CAA, respectively, over $n = 3$ biologically independent experiments for all conditions except for CGXII + PYR and CGXII + GLU where $n = 2$ and $n = 4$, respectively, biologically independent experiments were performed. **b** The ribosome densities from exponentially growing cells, shown in (**a**), are plotted against the growth rates of the respective cultures (black diamonds). The ribosome density

of stationary cells was included as a reference for zero growth rate (grey square). Shown is the mean and standard deviation for conditions Chemostat 0.2 h⁻¹, CGXII + PYR, Chemostat 0.3 h⁻¹, CGXII + PYR + CAA, Chemostat 0.43 h⁻¹, CGXII + GLU, CGXII + GLU + CAA, BHI + GLU and BHI + GLU + CAA for the exponentially growing cells and BHI + GLU for the stationary growth sample. The number of cells analysed and independent experiments is the same as indicated for Figs. 2b and 3a. **c** Comparison of the R/P ratio measurements (grey circles) to the SMLM data shown in panel b. Shown are mean values and standard deviations for the RNA/protein values for the following growth rates: 0.94 h⁻¹ ($n = 2$); 0.77 h⁻¹ ($n = 2$); 0.61 h⁻¹ ($n = 3$); 0.47 h⁻¹ ($n = 5$), 0.40 h⁻¹ ($n = 3$); 0.29 h⁻¹ ($n = 3$); 0.27 h⁻¹ ($n = 3$); 0.24 h⁻¹ ($n = 3$); 0.12 h⁻¹ ($n = 3$); 0.07 h⁻¹ ($n = 3$); 0.0 h⁻¹ ($n = 2$), where $n$ is the number of biologically independent samples analysed. Standard deviations were omitted for the SMLM data set to improve clarity. A segmental linear regression was applied to the data. Source data are provided as a Source data file.

(Supplementary Dataset 1) for 30–281 individual cells. As already described for other organisms, our results showed that also for *C. glutamicum*, the faster the growth rate the larger the cells are[33,37] and the more ribosomes they contain[7,26] (Fig. 2c, Supplementary Fig. 3, Supplementary Dataset 1). The observed mean increase in cell volume (from ~0.9 to ~2.2 μm³) at faster growth rates is mostly due to an increase in cell length from ~1.7 to ~2.7 μm (Fig. 2d, Supplementary Dataset 1), which results in the so-called surface area to volume ratio homeostasis[38]. The total emitter counts/cell (approximate number of ribosomes/cell) varied from about 12,000 to over 60,000 ribosomes/cell (Supplementary Fig. 3, Supplementary Dataset 1). There was a large variation in ribosome numbers/cell for cells cultivated in a certain medium, especially for the higher growth rates, possibly reflecting the heterogeneity and asynchrony of the culture (Fig. 2b, c, Supplementary Fig. 3, Supplementary Dataset 1).

To determine ribosome numbers independent of cell size, the mean ribosome densities, i.e. the number of ribosomes/μm³ were calculated. In exponentially growing cells, as growth rate increased, the mean ribosome density increased non-linearly from ~14,000 at μ = 0.2 h⁻¹ to ~30,000 at μ = 0.48 h⁻¹, whereas in stationary phase cells it was rather constant between 13,000 and 16,000, irrespective of the medium used for cultivation and the growth rate observed during the exponential phase (Fig. 3a, Supplementary Dataset 1). Thus, the mean ribosome density in stationary cells was very similar to that of cells growing exponentially at slow growth rates (Fig. 3a, Supplementary Dataset 1). In control SMLM experiments, a reversely labelled *C. glutamicum* SM55 strain was used containing uS2-EYFP and bL19-PAmCherry, where in this case bL19-PAmCherry fluorescent molecules were enumerated and taken as proxy for the amount of 50S ribosomal particles. Strain SM55 showed comparable ribosome densities as the SM34 strain under three different growth conditions (Supplementary Dataset 1), supporting the validity of the results.

To further check the accuracy of our SMLM-counting data, we estimated the number of ribosomes per cell from total RNA measurements and the mean colony-forming units (cfu) per OD₆₀₀ for two of the growth conditions previously used. In exponentially growing cells the majority (~86%) of the total RNA in the cell is rRNA[7,12,17]. The theoretical number of ribosomes can thus be calculated from the total RNA quantification from a known number of cells (calculated by determining the number of cfu OD₆₀₀⁻¹ mL⁻¹) and the rRNA mass of a single ribosome (2.53 × 10⁻¹⁵ mg; Supplementary Note 2). For cells growing exponentially in BHI + GLU medium and CGXII + GLU medium, 43,323 and 21,357 ribosomes/cell were calculated with this method, which is in very good agreement with the numbers determined by SMLM (45,207 and 25,770 ribosomes/cell).

## Ribosome quantification at the population level

The SMLM data revealed a mean ribosome density of ~15,000/μm³ up to a growth rate of 0.2–0.3 h⁻¹, which increased to ~30,000/μm³ at growth rates of ~0.5 h⁻¹ (Fig. 3b). To further confirm these results and exclude a possible bias in the ribosome quantification method used, we performed classical RNA-protein (R/P) ratio measurements. The R/P ratio is a surrogate and widely used method for the determination of ribosome abundance, since in exponentially growing cells the protein content remains largely invariant with the growth rate and the number of ribosomes is proportional to the rRNA, which comprises the majority of the total RNA of the cell[7,10,12,39,40]. To determine the R/P ratio, *C. glutamicum* wt was grown in batch culture in different media as for the SMLM measurements to cover a range of growth rates and cells were harvested during mid-exponential phase. Unlike *E. coli* or *Bacillus subtilis* that can reach growth rates >2 h⁻¹, *C. glutamicum* wt exhibits a maximal growth rate of ~0.6–0.67 h⁻¹ [41]. In order to extend the range of growth rates, an evolved *C. glutamicum* strain (EVO5)[42] was included for the R/P ratio measurements (Fig. 3c, Supplementary

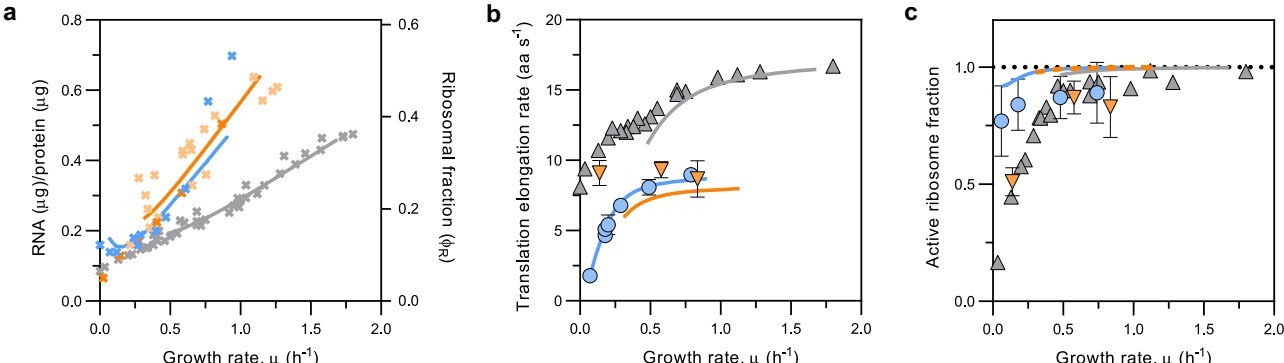

**Fig. 4 | Growth-rate dependency of ribosomal fraction, translation elongation rate, and active ribosome fraction of *C. glutamicum* differs from that of *E. coli* cultivated at 30 °C or 37 °C. a** R/P ratio and ribosomal fraction of *C. glutamicum* (blue) compared to that of *E. coli* cultivated at either 30 °C (orange) or 37 °C (grey). The data points for *E. coli* shown in dark orange were experimentally determined in this study, the others for 30 °C (light orange) were taken from literature[79,80], as well as those for 37 °C[4,7,16,21]. **b** Translation elongation rate for *C. glutamicum* (blue) and for *E. coli* 30 °C (orange, this study) and 37 °C (grey, extracted from ref. 21) growing at various growth rates. Depicted are mean values and standard deviation for the translation elongation rate for following growth rates: 0.79 h$^{-1}$ ($n$ = 3); 0.49 h$^{-1}$ ($n$ = 2); 0.29 h$^{-1}$ ($n$ = 2); 0.20 h$^{-1}$ ($n$ = 3); 0.18 h$^{-1}$ ($n$ = 2); 0.18 h$^{-1}$ ($n$ = 3); 0.07 h$^{-1}$ ($n$ = 2)

for *C. glutamicum* and 0.84 h$^{-1}$ ($n$ = 2); 0.58 h$^{-1}$ ($n$ = 2); 0.14 h$^{-1}$ ($n$ = 4) for *E. coli* where $n$ is the number of biologically independent repeats performed for each condition. Note that *C. glutamicum*, but not *E. coli*, shows very low translation elongation rates at low growth rates. **c** Active ribosome fraction calculated for different growth rates (Supplementary Note 4) for *C. glutamicum* (blue) compared to that of *E. coli* cultivated at 30 °C (orange, this study) or 37 °C (grey, extracted from ref. 21). Shown is the mean and standard deviation values. Please refer to the Supplementary Data 4 and 5 for the number of biologically independent repeats performed. Note that, in contrast to *E. coli*, *C. glutamicum* retains a high fraction of active ribosomes at low growth rates. Lines in (**a**), (**b**) and (**c**) were derived from model simulations. Source data are provided as a Source data file.

Dataset 2), which can reach mean growth rates up to 0.94 h$^{-1}$. Comparison of the R/P ratios, which varied from 0.14 to 0.7 µg RNA/µg protein, to the SMLM results showed a good agreement between the two methods (Fig. 3c). Importantly, both methods revealed a linear correlation between ribosome abundance and growth rate only above rates of ~0.4 h$^{-1}$. In cells with growth rates between 0 (stationary phase) and 0.3 h$^{-1}$, the ribosome abundance varied only to a low extent between 13,000 and 16,000 ribosomes/µm$^3$ or 0.14 and 0.19 µg RNA/µg protein (Fig. 3c).

**Translation elongation rate and active ribosome fraction**

A deviation from the linear relationship between R/P ratio and µ has also been reported for *E. coli* grown at 37 °C and growth rates <0.7 h$^{-1}$ [21], but it is much weaker than the one observed for *C. glutamicum* (Fig. 4a). In *E. coli*, the deviation is associated with a strong reduction of the active ribosome pool, hypothesized by Dai et al.[21] to be caused by either hibernation mechanisms[43], ppGpp-regulated translation initiation inhibition[44], or by passive translation abortion caused by ribosome stalling due to a decreased availability of translation precursors[45]. All ribosomes not participating in the translation process or unable to finish translation and yield the encoded protein are then considered inactive[21]. By reduction of the number of active ribosomes *E. coli* allows the preservation of high translation elongation rates during slow growth[21]. To determine if such a process also occurred in *C. glutamicum*, translation elongation rates and derived active ribosome fractions were measured for different growth rates using a fluorescence-based assay (see 'Methods' and Supplementary Notes 3 and 4). For this purpose, we followed the synthesis of EYFP as fluorescent reporter protein after IPTG induction and translation was stopped by chloramphenicol addition at short time intervals. Determination of the time at which fluorescence first appears allows for the calculation of the translation elongation rate. To calculate the time needed for translation initiation and also to confirm the translation elongation results obtained with the EYFP-reporter (238 amino acids), the larger reporter protein construct mCherry-L-EYFP (508 amino acids) was additionally used for some of the conditions tested (Supplementary Dataset 3). The translation elongation rate was calculated by dividing the number of amino acids (aa) for either construct by the time at which the first fluorescent molecules were

detected, subtracted by the time required for translation initiation. The translation elongation rate obtained for both constructs when grown under the same growth conditions was very similar and varied between 1.79 and 8.98 aa s$^{-1}$ (Supplementary Dataset 3). While in *E. coli* the translation initiation time is constant for cells growing at different growth rates and temperatures (~10 s)[21,46], in *C. glutamicum* the time for translation initiation was found to be ~10–12 s for all conditions tested. The translation initiation time was calculated, as previously described[21,46], from the time at which fluorescence first appears ($t_0$) for each of the two reporter constructs (EYFP, 238 aa or mCherry-L-EYFP, 508 aa) followed by the subtraction of the time needed for translation elongation (Supplementary Dataset 3). Assuming that the translation initiation is similar for both protein constructs when cells are growing in the same cultivation conditions, we can then estimate the translation elongation rate given by $k_{mCherry-L}$ = 270/($t_{0(mCherry-L-EYFP)}$ − $t_{0(EYFP)}$). Finally, the translation initiation can be calculated from $t_{initiation}$ = $t_{0(EYFP)}$ − 238/[270/($t_{0(mCherry-L-EYFP)}$ − $t_{0(EYFP)}$)] in which the second term is the time needed for the EYFP elongation.

The maximal translation elongation rate determined for *C. glutamicum* was ~9 aa s$^{-1}$ (Fig. 4b). This value is approximately half of 17 aa s$^{-1}$ determined for *E. coli* at 37 °C[18,21], but the same as reported in literature for *E. coli* at 30 °C[47] and as determined in this study (Fig. 4b and Supplementary Dataset 4). In order to support a certain growth rate at 30 °C, cells require more ribosomes than at 37 °C due to the lower translation elongation rate, which follows an Arrhenius kinetics for growth temperatures between 23–44 °C[47]. This is well reflected in the steeper slopes of the linear part of the Rb/µ correlation (Fig. 4a)[10]. Such steeper slopes are also observed in slow-translation mutants when compared to the parental strain[10].

In contrast to *E. coli* grown either at 30 °C (Supplementary Dataset 4) or 37 °C[21,48], the translation elongation rates of *C. glutamicum* reached very low values as the growth rate decreased (Fig. 4b and Supplementary Fig. 7b). In *E. coli* it is observed that during slow growth relatively high translation rates are maintained while the fraction of active ribosomes is reduced to under 20%[21] (Fig. 4c). In *C. glutamicum*, however, the fraction of active ribosomes remained above ~70% for all conditions tested (Fig. 4c). The higher fraction of active ribosomes

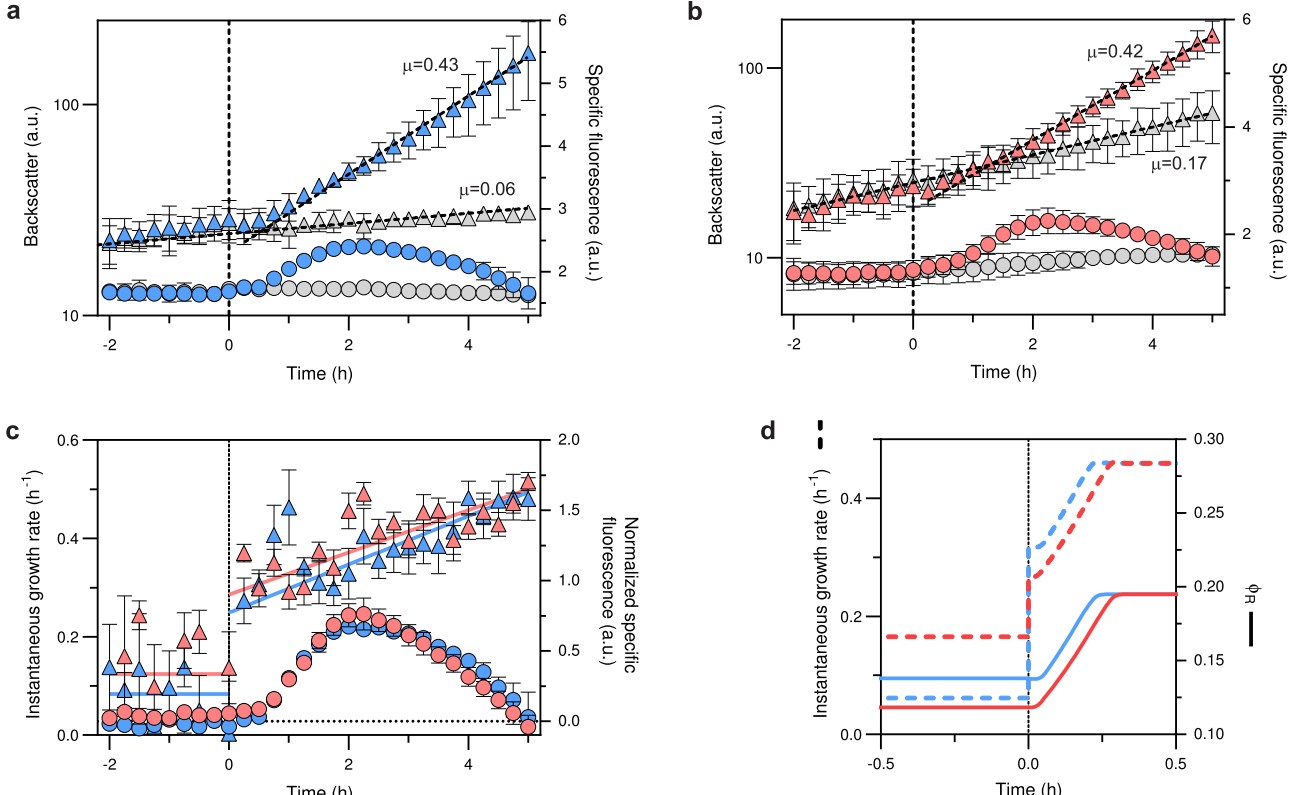

**Fig. 5 | Nutrient upshift results in an immediate increase in growth rate.** In the upshift experiments, *C. glutamicum* strain SM34 was initially grown in either CGXII +Glutamate (**a**) or CGXII+EtOH (**b**). At $t = 0$ h glucose was added (blue and red symbols) or the same volume of water was added to control wells (grey symbols). Backscatter (triangles) and specific EYFP fluorescence were followed online throughout the cultivation in a BioLector system. Ribosome abundance is reflected by the specific fluorescence for each time point (circles). Growth rates μ (h⁻¹) are indicated next to the growth curves and were determined by fitting a linear regression to the relevant backscatter values (discontinuous line). In panels (**a**) and (**b**), the mean and standard deviation of the backscatter values and the specific fluorescence are shown ($n = 3$ biologically independent experiments). **c** The instantaneous growth rates (triangles) were calculated from the upshift

experiments shown in (**a**) and (**b**). The solid lines show the mean instantaneous growth rate for values < 0 h and a linear regression fit for values > 0 h. The normalized specific fluorescence (circles) was calculated by subtracting the specific fluorescence of control wells without glucose from that of glucose-containing wells. Shown are the mean and SEM values of the instantaneous growth rate ($n = 3$ biologically independent experiments). **d** Upshift experiment simulation (see Methods, Supplementary Data 10). Shown are time courses of the instantaneous growth rate μ (dashed lines) and the ribosomal fraction $\Phi_R$ (solid lines, fraction of total ribosomal protein per total protein (w/w)) before and after increasing the nutrient quality parameter ($k_n$) from $k_{n,glutamate}$ (blue) or $k_{n,EtOH}$ (red) to $k_{n,glucose}$ (Supplementary Data 9). Source data are provided as a Source data file.

probably results in the depletion of precursors needed for translation and in a consequent decrease of the elongation rates and an increase in the time needed for translation initiation observed for *C. glutamicum* at very low growth rates (Supplementary Dataset 3). These results suggest that active ribosome inactivation e.g., via hibernation mechanisms[21], may not be as relevant for *C. glutamicum* as for *E. coli* under the conditions tested.

### Growth rate and ribosome abundance before and after a nutrient upshift

To exclude that the high abundance of ribosomes as well as the marked decrease in the translation elongation rate observed at slow growth rates for *C. glutamicum* was due to the accumulation of defective or misassembled ribosome particles, nutrient upshift experiments were performed. In the presence of defective ribosomal particles, we expected to observe an initial phase after the upshift dominated by the synthesis of functional ribosomes and consequently a delay in the increase of the instantaneous growth rate. On the contrary, if the ribosomes present in the cell are functional, they should allow for an immediate increase in growth rate. For this purpose the SM34 strain was used, enabling simultaneous measurement of growth (as backscatter) and ribosome production (as fluorescence) in a BioLector cultivation system.

Cells were initially grown in a poor carbon and energy source, using either glutamate or ethanol. Once cells reached the exponential growth phase, glucose was added as a good carbon source (Fig. 5a, b, $t = 0$, vertical dashed line).

Our results show that immediately after glucose addition there is a very rapid initial jump in the instantaneous growth rate to ~0.3 h⁻¹, suggesting that the excess ribosomes present are functional, followed by a second slower phase where ribosomes are produced and accumulate to support translation at post-shift rates as evidenced by the progressive increase in specific fluorescence (Fig. 5c) observed ~30 min after the upshift. EYFP has a maturation time of $34.0 \pm 4.3$ min in live *E. coli* cells cultivated at 32 °C[49]. Therefore, ribosome production probably started very soon after glucose addition. Despite different initial growth rates on glutamate and ethanol (0.06 h⁻¹ and 0.14 h⁻¹, respectively) ribosome accumulation followed very similar kinetics (Fig. 5c). These results show that in both conditions there is a pre-existing ribosome overcapacity that allows cells to instantly respond to the improved growth conditions shown by the immediate increase in the instantaneous growth rate to 0.3 h⁻¹. In order to support growth rates above 0.3 h⁻¹ cells need to synthesize additional ribosomes, as evidenced in Fig. 5c, and supported by the linear correlation between growth rate and ribosome abundance observed for growth rates >0.4 h⁻¹ (Fig. 3c).

**Mechanistic insights into the Rb/μ correlation in *C. glutamicum***

To gain further insights into the dependency between growth rate and ribosomal (mass) fraction ($\Phi_R$, ribosomal proteins/total proteome, see also Supplementary Note 5A), and, in particular, to understand if depletion of translation precursors could explain the very low translation rates observed during slow growth, a coarse-grained self-replicator (SR) model, previously formulated for *E. coli* at μ > 0.5 h⁻¹,⁵⁰, was re-calibrated with data collected for *C. glutamicum* at different growth rates/conditions (see Methods, Supplementary Note 5C, Supplementary Dataset 9). SR models link the molecular composition of cells with their growth rate, using few macro-reactions, on the basis of resource re-allocation principles, at steady-state and under nutrient shifts[50,51]. For a given nutritional state, described by a single (phenomenological) nutrient supply quality parameter ($k_n$), the species-specific SR models are used to infer translation elongation ($k_R$) and ppGpp-regulated ribosome synthesis ($v_{rrn}$) rates from experimentally determined sets of ribosomal fractions ($\Phi_R$) and corresponding specific growth rates (μ). In the models, translation inefficiency, i.e. the amount of uncharged tRNA-ribosome complexes, is used as a trigger for the regulation of ribosome synthesis via a ppGpp-based regulatory mechanism.

Equipped with this simple regulatory mechanism, the SR model for *C. glutamicum* accurately described the nonlinear Rb/μ correlation over the entire range of growth rates examined in this work (μ = 0.07–0.94 h⁻¹, Fig. 4a and Supplementary Note 6 for comparison with *E. coli*). From these simulations, we inferred key rates of translation elongation and ribosome production, i.e. the maximum translation elongation rate ($k_{R,max} = 9.3 \pm 0.8$ aa ribosome⁻¹ s⁻¹) and the *rrn* operon transcription initiation rate ($v_{rrn} = 0.3 \pm 0.1 – 11.1 \pm 3.8$ *rrn* initiations s⁻¹ cell⁻¹). The estimated $k_R$ values are in excellent agreement with the experimentally determined values (Fig. 4b), whereas $v_{rrn}$ rates are in the same order of magnitude as those calculated from cellular ribosome numbers (Supplementary Fig. 7c). Furthermore, from the modelling results, the active ribosome fraction was predicted to be high (91.7–99.8% of the total ribosomes) for the entire range of growth rates (Fig. 4c). These high fractions of active ribosomes (Supplementary Fig. 8c) are consistent with the observation of high amounts of charged tRNAs (Supplementary Fig. 8a). Those were estimated to be at least two orders of magnitude higher than concentrations of inactive ribosomes (Supplementary Fig. 8d), i.e. uncharged tRNA-ribosome complexes (Supplementary Fig. 8b). Towards low growth rates, the ppGpp concentration (Supplementary Fig. 8e) steeply rises in response to an increase in the amount of uncharged tRNA. To summarize, results not only predict the maintenance of a high fraction of active ribosomes over the whole range of growth rates, but also the decrease of translation elongation rates due to depletion of translation precursors (such as charged-tRNAs) at slow growth (Supplementary Fig. 8a), analogous to what was experimentally observed (Fig. 4b, c). Therefore the coarse-grained model with inbuilt regulatory mechanism for control of ribosome synthesis was able to reproduce the experimental data for *C. glutamicum* over the whole range of observed growth rates. This finding suggests that in these growth regimes no ribosomal hibernation mechanism deliberately inactivates a proportion of the active ribosomes in favour of increased translation elongation rates as it is known for *E. coli*[21]. Noteworthy, this sets a counterpoint to *E. coli*, where the corresponding SR model for *E. coli* expectedly cannot explain the relatively high translation rates during slow growth (30 °C: μ < 0.3 h⁻¹ and 37 °C: μ < 0.5 h⁻¹, Supplementary Note 6), as the model does not include a hibernation mechanism.

To further analyse the response to nutrient upshifts, we modelled the adaptation of *C. glutamicum* when shifted from a poor to a good carbon source (Fig. 5d). In the upshift simulations, the initial progressive increase in ribosome fractions observed in the experimental results (Fig. 5c) was well captured (Fig. 5d) as result of the expected rise in ribosome production rate ($v_{rrn}$, Supplementary Fig. 9a). In the simulation Supplementary Fig. 9a, $v_{rrn}$ initially overshoots before

leveling off at a new steady-state value. Furthermore, transition into a richer medium resulted in increased ribosome activity (Supplementary Fig. 9b) due to the presence of the additional translation precursors. The latter caused a jump in translation rate ($k_R$) (Supplementary Fig. 9a) and a consequent step increase in growth rate (Fig. 5c, d), likely modulated by the intracellular ppGpp-concentration (Supplementary Fig. 9c). Since no mechanism for ribosome inactivation was implemented in the model, a direct comparison of in silico upshift experiments between *C. glutamicum* and *E. coli* at 30 °C was only possible at growth rates μ ≥ 0.30 h⁻¹. Comparison of in silico upshift experiments of both strains to the same post-shift medium quality at approx. μ = 0.3 h⁻¹ yielded qualitatively and quantitatively similar results (Supplementary Fig. 9d–g), which was expected because the calibrated models of *E. coli* and *C. glutamicum* differed only in the two estimated parameters $k_{R,max}$ and $v_{rrn,max}$, which were estimated to be 1.1-fold increased and 3.2-fold decreased in *C. glutamicum* compared to *E. coli* (Supplementary Dataset 7). Supplementary Fig. 9d shows that the immediate growth rate of *E. coli* after the shift was ~20% higher compared to *C. glutamicum* (μ = 0.39 h⁻¹ and 0.32 h⁻¹, respectively), which is explained by the increased ribosomal fraction (+29% in *E. coli*, Supplementary Fig. 9d), while the translation elongation rate was slightly decreased (−8% in *E. coli*, Supplementary Fig. 9e) before the shift. Similar pre- and post-shift growth rates were experimentally observed for *E. coli* at 30 °C (μ = 0.27 and 0.39 h⁻¹, respectively)[6]. Furthermore, *E. coli* reached the new steady-state growth rate faster because the transient ribosome production rate was approx. three times higher at its peak (Supplementary Fig. 9e, Supplementary Note 6).

Finally, to investigate the maximum possible growth rate of *C. glutamicum*, e.g. after a nutrient upshift to a "super-rich" medium, extrapolation of the nutrient quality $k_n$ (Supplementary Note 5D) predicted a global maximum specific growth rate $\hat{\mu}_{max} = 0.94$ h⁻¹, which is very close to the experimentally determined mean growth rate of the laboratory-evolved EVO5 strain[42] (Supplementary Dataset 2).

## Discussion

In this study, we examined two fundamental physiological properties of *C. glutamicum*, the ribosome content and the translation elongation rate over a range of growth rates. We also analysed the ribosome localization and observed that the ribosomes are preferentially located outside the nucleoid areas during exponential and stationary growth phases. Compartmentalization of translation can have important physiological implications. The organization of chromosomal DNA into nucleoids within the cell generates DNA-free areas where the mobility of large complexes, such as ribosomes, is found to be increased when compared to that of bacterial species where no separation between DNA and ribosomes is observed[33]. Furthermore, it can influence mRNA localization, either close to the gene locus from which the mRNA was transcribed from ref. 52 due to decreased local diffusion or close to the cellular location of the encoded protein(s)[53,54]. We speculate that the latter may also occur in *C. glutamicum* since the ribosomes are preferentially found in nucleoid-free areas.

To quantify ribosomes in single cells, we developed a SMLM method together with the emitter counter software *SurEmCo*. This method and the analysis tool may easily be applied to the study of other proteins in this or other organisms. From our estimates, a resting *C. glutamicum* cell in stationary phase possesses ~15,000 ribosomes/μm³. At the highest growth rate of ~0.5 h⁻¹ analysed by SMLM, this number doubles to ~30,000 ribosomes/μm³, which is close to the reported 27,000 ribosomes/μm³ for *E. coli* at 30 °C and μ ≈ 0.7 h⁻¹ determined by a different super-resolution microscopy method[26]. However, in our SMLM analysis we remarked a steep increase in ribosome abundance only above μ ≈ 0.4 h⁻¹ and these results were confirmed by R/P ratio measurements of the wt and an evolved faster-

growing *C. glutamicum* strain (Fig. 3c). Such marked deviation of R/P ratio from linearity during slow growth has also been reported for *Aerobacter aerogenes*[12]. In *E. coli*, although present, this deviation is considerably more moderate[21]. The inflection point at which the R/P ratio starts to linearly rise with the growth rate is approximately the point at which the translation elongation rate in *C. glutamicum* is close to its maximum value. At μ between 0.07–0.29 h$^{-1}$, faster growth of *C. glutamicum* is thus mainly achieved by the observed progressive increase in the rate of translation elongation, from 1.79 to 6.79 aa s$^{-1}$, rather than by an increase in ribosome abundance, which remains constant for μ < 0.4 h$^{-1}$. Unlike *E. coli* 37 °C, for which the translation elongation rate varies only by approximately 2-fold between slow and maximal growth rates[20,21], for *C. glutamicum* we observed a five-fold increase in the translation elongation rate across all steady-state growth conditions at which it was measured (1.79–8.98 aa s$^{-1}$). The marked decrease in translation elongation rate at low growth rates was explained by a calibrated coarse-grained self-replicator model that uses the amount of uncharged tRNA-ribosome complexes as signal for the regulation of ribosome synthesis.

For *E. coli* at 30 °C, we measured a constant peptide elongation rate of ~9 aa s$^{-1}$ at three different growth rates (0.14, 0.58 and 0.84 h$^{-1}$). Dai et al.[31] show that the translation elongation rate in *E. coli* is kept high due to substantial ribosome inactivation during slow growth. However, for *C. glutamicum* we observe that the fraction of active ribosomes is instead kept high, above ~70% for all tested conditions. Ribosome hibernation mechanisms, in which bacteria inactivate a fraction of their own ribosomes to reduce the cost of protein production while waiting for more favourable conditions, have been elucidated for a few organisms[43]. In *C. glutamicum*, homologues of proteins shown to be involved in either ribosome inactivation, such as MPY (mycobacterial protein Y similar to long hibernation promoting factors, cg0867) and EttA (energy-dependent translational throttle A, cg2695), or in modulation of ribosome activity, such as RsfS (ribosome silencing factor, cg2582) can be found. The efficacy and importance of each of these mechanisms for ribosome inactivation in *C. glutamicum* needs to be elucidated. However, the presence of a large fraction of active ribosomes argues against the relevance of such mechanisms under the conditions examined in this study.

Ribosome biogenesis is tightly controlled in a growth rate-dependent manner[55]. The alarmone (p)ppGpp is a major modulator of the bacterial growth rate, as it regulates ribosome production in response to the nutritional state of the cell[56]. In *C. glutamicum* most translation-associated genes are transcriptionally downregulated in a (p)ppGpp-dependent manner[57]. However, in contrast to other bacteria such as *E. coli*, where accumulation of uncharged tRNAs strongly activates ppGpp synthesis, in *C. glutamicum* and in mycobacteria strong ppGpp upregulation is only observed upon complete depletion of all carbon-, nitrogen- and phosphorus-containing nutrients[57–59]. Differences in the stringent response triggers have been suggested to reflect adaptations to the distinct lifestyle of each species[60]. Excess ribosomes present during slow growth have also been described for other bacteria[61–64] and archaea[65] as well as a significant reduction of the translation elongation rate[63,65], which can be seen as evolutionary adaptations. In conclusion, maintenance of an excess of slow-working ribosomes under poor nutrient conditions may allow for a large translational overcapacity that enables quick growth recovery of *C. glutamicum*. This strategy is different from the one employed by *E. coli* or yeast[21,66], where towards low growth rates the ribosomal fraction and in particular the fraction of active ribosomes is reduced to enable a high translation elongation rate. It is not clear what the advantage of adopting one over the other strategy is, however, it may become relevant and provide a competitive advantage during quick growth recovery in the respective environmental niches.

## Methods

### Bacterial strains and plasmids

The strains used in this study were the *C. glutamicum* type strain ATCC 13032 (wt) or its derivatives MB001(DE3), a prophage-free strain, which allows for IPTG-inducible T7-RNA polymerase-dependent gene expression[67] that shows a very similar growth behaviour as the wt[68], and EVO5, a laboratory-evolved faster-growing strain[42]. To measure the translation elongation rate, the *C. glutamicum* MB001(DE3) strain carrying the reporter plasmid pMKEx2-*eyfp*[67] or pMKEx2-*mCherry-linker-eyfp* was used. *E. coli* DH5α was used in all cloning procedures and the JM109(DE3) strain was used for the determination of R/P ratios as well as translation elongation rates.

For fluorescent co-imaging of ribosomes and nucleoids, *C. glutamicum* wt was modified via homologous recombination to carry a translational fusion between the ribosomal protein bL19 (RplS) and the EYFP fluorescent protein, giving rise to strain SM30. The *eyfp* gene was amplified from plasmid pEKEx2-*eyfp*[69]. It originates from pEYFP-C1 (Clontech) and encodes an EYFP protein of 238 amino acids with two amino acid changes (D130N and D174G). The *rplS* (cg2235) gene along with 1 kb of its upstream region, the EYFP coding gene, and 1 kb of the *rplS* immediate downstream region were PCR-amplified. The resulting DNA fragments were subsequently cloned in the above indicated order (1kb_upstream-*rplS*-linker-*eyfp*-1kb_downstream) via Gibson assembly into the EcoRI-digested pK19*mobsacB C. glutamicum* suicide vector[70]. During cloning the linker sequence 5′-GGCGCTGCTGCTGCTGGC-3′ corresponding to the amino acid sequence GAAAAG was inserted between *rplS* and *eyfp* gene sequences. The resulting plasmid was introduced into *C. glutamicum* wt via electroporation. After selection for genomic integration of the plasmid on kanamycin-containing plates, the second homologous recombination event was enforced by cultivation on sucrose-containing plates. Finally, several kanamycin-sensitive and sucrose-resistant clones were checked for correct integration using oligonucleotides annealing outside the modified region and Sanger sequencing of the region of interest.

For SMLM, strain SM30 was modified via homologous recombination to carry an additional translational fusion, in this case between the ribosomal protein uS2 (RpsB) and the photoactivatable mCherry fluorescent protein (PAmCherry)[71], yielding strain SM34. To construct the uS2-PAmCherry translational fusion, the *rpsB* (cg2222) gene along with 1 kb of its upstream region, the PAmCherry encoding gene and 1 kb of the *rpsB* immediate downstream region were amplified by PCR. As above, the resulting DNA fragments were subsequently cloned in the presented order (1kb_upstream-*rpsB*-linker-*PAmCherry*-1kb_downstream) via Gibson assembly into the EcoRI-digested pK19*mobsacB* suicide vector[70]. During cloning the linker sequence 5′-CAG-GAAAGGCGACAGGAG-3′ corresponding to the amino acid sequence QERRQE[26] was inserted between *rpsB* and *PAmCherry* gene sequences. Double crossover selection and correct integration verification was performed as described above for SM30. To construct strain SM55, which carries the translational fusions uS2-EYFP and bL19-PAmCherry, a similar approach as described above for SM34 was followed. In this case, we started by creating a strain containing the bL19-PAmCherry translational fusion (linker sequence 5′-GGCGCTGCTGCTGCTGGC-3′ encoding the amino acid sequence GAAAAG) into which the uS2-EYFP fusion (linker sequence 5′-CAGGAAAGGCGACAGGAG-3′ encoding the amino sequence QERRQE) was subsequently introduced.

### Growth medium and cultivation conditions

*C. glutamicum* strains were cultivated either in brain heart infusion (BHI) medium (Bacto™, BD, Heidelberg, Germany) or in CGXII mineral medium containing per liter of distilled water: 1 g K$_2$HPO$_4$, 1 g KH$_2$PO$_4$, 10 g (NH$_4$)$_2$SO$_4$, 5 g urea, 21 g 3-(*N*-morpholino)propanesulfonic acid (MOPS), 0.25 g MgSO$_4$•7H$_2$O, 10 mg CaCl$_2$, 0.2 mg biotin, 10 mg FeSO$_4$•7H$_2$O, 10 mg MnSO$_4$•H$_2$O, 1 mg ZnSO$_4$•7H$_2$O, 0.3 mg CuSO$_4$, 0.02 mg NiCl$_2$•6H$_2$O, supplemented with 30 mg of

3,4-dihydroxybenzoate (iron chelator) and a carbon source (below). The pH was adjusted to 7.0 with KOH. The different carbon sources were provided at the following final concentrations: 2% (w/v) glucose (GLU), 2% (w/v) sodium acetate (ACE), 2% (w/v) sodium pyruvate (PYR), 2% (w/v) sodium DL-lactate (LAC), 1.1% (v/v) ethanol (EtOH), and 0.9% (w/v) sodium glutamate (Glutamate). In a few conditions 0.2% (w/v) casamino acids (CAA) or yeast extract (YE) was added. Whenever necessary, the medium was supplemented with kanamycin (25 μg/mL).

*C. glutamicum* cultures were always cultivated at 30 °C and followed three steps: seed culture, preculture, and main culture. In the seed culture, single clones were grown overnight (ON) in BHI + GLU and then diluted 1:100 for the precultures, which were grown for ~18 h in the same medium as the experimental cultures. Finally, the experimental cultures were inoculated at a normalized starting optical density at 600 nm ($OD_{600}$) of 0.5 or 1.0. For microscopy, cells were grown in 10 mL of culture medium in 100 mL baffled flasks shaken at 130 rpm. For RNA/protein measurements, cells were grown in 50 mL of culture medium in 500 mL baffled flasks shaken at 130 rpm or in a BioLector cultivation system (m2p-labs, Baesweiler, Germany). For BioLector microcultivations, cells were grown in 800 μL culture volume in a 48-well FlowerPlate (m2p-labs, Baesweiler, Germany) shaken at 1200 rpm.

To control for environmental variations such as the supplied medium as well as those inherent to shake flask cultivations, such as oxygen availability and/or pH shifts, samples were also taken from continuous chemostat cultivations. For the chemostat cultivations, a 3 L steel bioreactor (KLF 2000, Bioengineering, Switzerland) equipped with process control (Lab view 2010, National Labs) was used. Continuous dilution was regulated with an advanced controller scheme[42] and three dilution rates were set to obtain exponential growth rates of $\mu = 0.2$, 0.3, and 0.43 $h^{-1}$. Cells of strain SM34 were cultivated in 1.2 L CGXII medium supplemented with 2% (w/v) glucose and 0.2% (w/v) CAA at 30 °C and 1.5 bar. pH was controlled at 7.4 via automatic addition of 25% (v/v) $NH_4OH$ solution and dissolved oxygen was kept above 70% by setting constant stirrer speeds of 700 rpm and aeration rates of 0.85 L $min^{-1}$. After steady state was reached for each of the set growth rates, samples were taken and the cells were immediately prepared for single molecule localization microscopy (see below).

*E. coli* was cultivated in LB or MOPS minimal medium plus 0.2 mM thiamine[21] supplemented with different carbon sources: 0.2% (w/v) glucose (GLU), 0.2% (w/v) casamino acids (CAA), 0.2% (v/v) glycerol (GLY), 20 mM sodium aspartate (ASP), or 20 mM sodium glutamate (Glutamate). *E. coli* was cultivated at 30 °C from a starting $OD_{600}$ of 0.03–0.05 in either 50 mL of culture medium in 500 mL baffled flasks shaken at 130 rpm for R/P ratio measurements or in 96-well plates with 200 μL culture medium in a Tecan Infinite M1000 Pro microplate reader (Tecan Group Ltd, Männedorf, Switzerland) with 216 rpm and 3 mm shaking orbital frequency and amplitude, respectively, for translation elongation rate determination.

## Upshift experiments
*C. glutamicum* strain SM34 was grown in either CGXII + EtOH or CGXII + Glutamate in a BioLector cultivation system. At time point zero ($t_0$), glucose was added to the medium to a final concentration of 2% (w/v) or an equal volume of water was added to control wells. Growth was followed by the increase in backscatter whereas ribosome production was observed by the fluorescent EYFP signal. Backscatter and fluorescence measurements were performed every 15 min. For each time point, instantaneous growth rates were determined from the backscatter measurements of adjacent points $\mu_x = (\ln B_{x+1} - \ln B_{x-1})/\Delta t$. Values of the simulated post-shift rates were obtained $-t_0 + 10$ s after changing the pre-shift steady state.

## Western blotting
Samples from mid-exponential *C. glutamicum* wt or SM34 cultures grown in BHI + 2% (w/v) glucose were lysed by bead beating

(Precellys24, Peqlab Biotechnologie, Erlangen, Germany) and cleared by centrifugation at 4 °C and $20,238 \times g$ for 10 min. The supernatant was collected and the protein concentration determined by the BC Assay Protein Quantitation Kit (Interchim, Montluçon, France). Protein samples (20 μg) were separated by SDS-PAGE (Mini-protean TGX any-kD, Bio-Rad, Hercules, CA, USA) and transferred onto a PVDF membrane (0.2 μm, Novex, Thermo Fisher Scientific Inc., Waltham, MA, USA). Membranes were blocked with 5% (w/v) non-fat dried milk in phosphate-buffered saline (PBS) containing 0.1% (v/v) Tween20 (PBST) for 30 min at room temperature. The membranes were then incubated with either primary monoclonal antibodies anti-mCherry (Takara Bio Cat# 632543, RRID:AB_2307319, Takara Bio USA, Inc., Mountain View, CA, USA) or anti-GFP (Antibodies-Online Cat# ABIN559689, RRID:AB_10852039, Clone GF28R, antibodies-online GmbH, Aachen, Germany), diluted 1:2000 and 1:3000 in PBST, respectively. After washing with PBST, the membranes were incubated with the anti-mouse secondary antibody conjugated to alkaline phosphatase (Sigma-Aldrich Cat# A3562, RRID:AB_258091, lot#107K6084, Sigma-Aldrich, Burlington, MA, USA) at a dilution of 1:5000 in PBST. Chemiluminescent signal detection was performed on a Fujifilm LAS-3000 imager (Fujifilm, Minato, Japan) after incubation with CDP-Star (Thermo Fisher Scientific Inc., Waltham, MA, USA).

## Wide-field and single-molecule localization microscopy
*C. glutamicum* SM30 was cultivated in BHI + GLU until mid-exponential phase ($OD_{600} \sim 3.0$–5.0) or until stationary phase (24 h). At these time points, 500 nM SYTOX Orange (ThermoFisher Scientific, Waltham, MA, USA) was added for DNA staining and the cultures were further incubated under the same conditions for 10 min. SYTOX Orange yielded the best results in terms of detected fluorescence versus background autofluorescence and was used as previously published for *E. coli* and *B. subtilis*[72]. Samples of 150 μL were collected by centrifugation at $5000 \times g$, washed once in 1 mL PBS and fixed with a 4% (w/v) paraformaldehyde solution in PBS. After 20 min incubation at room temperature the reaction was stopped by centrifuging and resuspending the cells in PBS containing 10 mM glycine. Cells were imaged on 1% (w/v) agarose pads containing PBS mounted on a glass slide suitable for imaging with an immersion objective through the glass (150–170 μm thickness).

All fluorescence microscopy measurements were performed with a self-constructed wide-field and TIRF fluorescence microscope with single-molecule sensitivity[35] based on an Olympus IX-71 inverted microscope body. It uses an AOTF (AOTF nC-VIS-TN 1001; AA Opto-Electronic, Orsay, France) to control the throughput of the two continuous wave laser sources that we used, i.e. an Argon-ion laser (514 nm; Coherent Innova 70 C, Coherent Inc., Santa Clara, US) for excitation of EYFP and a 561 nm solid state OPSL laser (Sapphire 561-200 CDRH-CP; Coherent Inc.) for excitation of photoactivated PAmCherry or SYTOX Orange. An Olympus ApoN 60x oil TIRF objective (NA 1.49) was used. Excitation and emission light were separated via a multiband dichroic mirror (Di01 R442/514/561; Semrock IDEX Health and Science LLC, Rochester, NY, USA) in combination with a multiple bandpass filter (FF01-485/537/627; Semrock) and an additional single bandpass filter, either FF01-609/57 (Semrock) for imaging SYTOX Orange and photoactivated PAmCherry or FF03-525/50 (Semrock) for EYFP imaging. Images were recorded with an EMCCD camera (Andor iXon DU897E-C00-#BV, Oxford Instruments, Abingdon, UK) cooled to −75 °C using a resolution of 512 × 512 pixels. The image from the microscope is additionally magnified via an achromatic lens (focal point, 50 mm; AC254-050-A-ML; Thorlabs, Bergkirchen, Germany). By adjusting (motorized) the lens and camera position, the pixel size can be adjusted between 65 and 130 nm/pixel. We used 80 nm pixel size. The camera acquisition time was set to 50 ms in all experiments. It is followed by a 35 ms readout time during which the camera cannot detect photons, i.e., in an image sequence, every image represents

85 ms, with an effective detection time of 50 ms. Wide-field images were calculated as the mean image of a short image series (typically 50) using 1 or 2% of the excitation laser output power.

For SMLM, cell samples from mid-exponential or stationary cultures were centrifuged at $5000 \times g$, washed once in 1 mL PBS and fixed with 4% (w/v) paraformaldehyde as detailed above. A volume of 200 µl of appropriate cell dilutions in PBS was applied to a well of an 8-well chambered micro-slide (µ-Slide 8 Well Glass Bottom, Cat.No: 80827 ibidi GmbH, Gräfelfing, Germany) previously coated with 0.1% (w/v) poly-L-lysine. After 30 min incubation at room temperature the micro-slide chambers were washed twice with PBS to remove any unattached cells. The slides were kept at 4 °C for a maximum of 7 days before use. SMLM was performed with the photoactivatable fluorescent protein PAmCherry[71]. PAmCherry cannot be excited at the imaging wavelength of 561 nm, since it absorbs in the near UV/blue spectral region (absorption maximum), fluorescing very dimly in the green spectral region. Photoactivation occurs when applying 405 nm light, upon which PAmCherry is transformed in a complex photochemical reaction[73] to a bright emitting fluorophore with an excitation maximum at 564 nm and an emission maximum at 595 nm. Super-resolved images were obtained by recording a long sequence of wide-field images (50 ms acquisition) with high power (75%) 561 nm laser light under continuous 405 nm diode laser exposure (Cube 405 - 100C; Coherent). As we intended to count all ribosomal PAmCherry fusion proteins present in the cells, we had to ensure that only a few photoactivated PAmCherry molecules per cell were present at a time and were detected as single molecules by excitation at 561 nm. For this reason, we used at the beginning of the SMLM acquisition a very low power of the photoactivating 405 nm light and manually increased the 405 nm laser power to keep a constant, low level of emitters per cell to avoid too many emitting PAmCherry molecules in single images. In fact, in some experiments there was a minute fraction of PAmCherry molecules that were already absorbing at 561 nm before the photoactivating light was applied. Therefore, we first acquired 100–300 single molecule images without any 405 nm light before we started the photoactivation.

Typically, we had 0 to at most 5 PAmCherry molecules per cell emitting at the same time, i.e., in the same wide-field image. In this way we ensured that we do not fail to detect any emitter due to problems of insufficient spatial separation of single molecules in single images as it would occur using SNSMIL or any other standard single molecule localization algorithms[74]. If emitters are too dense and their diffraction-limited representations (circles with diameter of ca. 250 nm) would overlap, SNSMIL would interpret the resulting fluorescence pattern as one single emitter (instead of two or three) and we would have systematically determined too low ribosome numbers.

As mentioned above, we gradually increased the 405 nm light power when recording the single molecule wide-field fluorescence image sequence, once the number of emitters in the current image was rather low (0 or 1 emitter per cell). We continued in this way until a further increase of the 405 nm light power did not generate new emitting single molecules. The recorded image sequences contained 5000–60,000 images and were analysed by SNSMIL, which determined the number, intensities and positions of all single molecule emitters in all cells and in all images of the image sequence. These data were further analysed by an in-house developed single molecule tracking programme to correct for two additional problems in the determination of ribosome number per cell (Supplementary Note 1).

Microscopy images were generated using Fiji(ImageJ).

## Ribosome counting by SMLM
To derive per cell ribosome count metrics, previously reconstructed snapshot SMLM data was tracked over time, and emitters assigned to individual cells. To this end, a custom software *SurEmCo* was developed (https://github.com/modsim/suremco). The Python software identified individual cells using a local thresholding method[75] applied to an acquired transmission image. Identified cell areas were used to select all raw emitter positions inside cells as generated by SNSMIL[36]. Emitters (i.e. ribosomes) were tracked throughout the image sequence per cell region, since they might slightly drift over time. Every emitter track with at most one dark frame in between was counted as one emitter to counteract potential biases in emitter counts caused by the presence of the same fluorophore in consecutive frames and the blinking probability of the fluorophore (Supplementary Note 1). Results were visualized in 3D, and metrics provided in a tabulated format. All processed outcomes were manually examined to exclude contributions originating from falsely identified cellular regions or superposed cells.

## RNA extraction
For each condition, the volume equivalent to an $OD_{600}$ of 3 or 5 per mL of a mid-exponentially growing culture was centrifuged in screw cap centrifuge tubes, the supernatant discarded, and the cells flash frozen in liquid nitrogen and stored at −80 °C. Pellets were resuspended in 500 µL QIAzol lysis reagent (Qiagen, Hilden, Germany) and incubated at room temperature for 1 min. Cells were disrupted using glass beads (3 × 30 s at 6500 rpm) in a Precellys 24 homogenizer (PEQLAB Biotechnologie GmbH, Erlangen, Germany) with a 5 min incubation on ice in between each disruption cycle. RNA was extracted once with chloroform using phase lock gel tubes (5 PRIME GmbH, Hilden, Germany), precipitated with isopropanol, and washed with 70% (v/v) ethanol. RNA pellets were allowed to dry at room temperature before resuspension in RNase-free water. RNA concentration was determined using the RNA ScreenTape Assay (Agilent Technologies Inc., Santa Clara, CA, USA) or by using the Qubit RNA BR Assay Kit (ThermoFisher Scientific), both according to the manufacturer's instructions. Both RNA and protein amounts refer to batch culture measurements from a "standard culture volume", e.g., 1 mL of culture normalized to an $OD_{600}$ of 1.

## Protein extraction
For each condition, the volume equivalent to an $OD_{600}$ of 3 or 5 per mL of a mid-exponentially growing culture was centrifuged in a screw cap centrifuge tube, the supernatant discarded, and the cells washed once in PBS, flash frozen in liquid nitrogen, and stored at −80 °C. Samples were thawed in 500 µL of PBS containing protease inhibitor cocktail cOmplete™ Ultra (Roche, Basel, Switzerland). Cells were disrupted as for the RNA extraction. Total protein content was determined using the BC Assay Protein Quantitation Kit (Interchim, Montluçon, France) according to the manufacturer's instructions.

## Estimation of the ribosome number per cell
*C. glutamicum* wt was grown in either BHI + GLU or CGXII + GLU until mid-exponential phase. Mean colony forming units (cfu) per $OD_{600}$ per mL were determined by plate enumeration of serial dilutions for two biological replicates, giving values of $2.39 \times 10^8$ (BHI + GLU) and $3.56 \times 10^8$ (CGXII + GLU). Total number of ribosomes was calculated according to $N_{Rb} = 0.86 \cdot R/m_{rRNA}$ (Supplementary Note 2) using the total RNA values (µg $OD_{600}^{-1}$ mL$^{-1}$) from Supplementary Dataset 2.

## Translation elongation rate assay
Translation elongation rates were measured via a fluorescent assay. *C. glutamicum* MB001(DE3) carrying either the expression plasmid pMKEx2-*eyfp*[67] or the longer construct pMKEx2-*mCherry-linker-eyfp* was cultivated in different media to yield a range of growth rates in a BioLector microcultivation system. One hour before induction of *eyfp* or *mCherry-linker-eyfp* expression with isopropyl-β-D-thiogalactoside (IPTG), the BioLector 48-well FlowerPlate was transferred to another BioLector instrument integrated into a robotics platform[76] and

incubated under the same conditions. At mid-exponential phase *eyfp* or *mCherry-linker-eyfp* expression was induced with 5.5 mM IPTG in triplicate. Immediately after induction, translation was stopped by adding 0.9 mg/mL chloramphenicol at 20–60 s intervals. The robotics platform was used as it allows accurate time intervals for both the addition of IPTG and chloramphenicol. In control wells, IPTG, chloramphenicol, or both compounds were substituted by the addition of an equivalent volume of cultivation medium. The BioLector FlowerPlate was further incubated at 30 °C ON to allow full maturation of the EYFP or mCherry-L-EYFP fluorescent proteins. Cell growth (backscatter at 620 nm) and fluorescence (excitation 508 nm, emission 532 nm) were monitored every 5 min. After background correction, the specific fluorescence for each well [(fluorescence$_{ti}$ − fluorescence$_{to}$)/(backscatter$_{ti}$ − backscatter$_{to}$)] was calculated for each time point. The mean specific fluorescence signal after fluorescence reached a constant maximum was used to create the time-dependent translation plots. A linear regression was fit to the data in GraphPad Prism 9.1.2 and used to determine the time point at which the first fluorescent molecules appeared (y = 0). The translation elongation was calculated by dividing the number of amino acids in EYFP (238) or mCherry-L-EYFP (508) by the determined time at which the first fluorescent proteins appear, from which the time for translation initiation (Supplementary Dataset 3) was subtracted and the 7 s delay inherent to the robotics setup was accounted for.

A similar approach was followed to measure the translation elongation rate in *E. coli* with the following modifications. The strain JM109(DE3) pMKEx2-*eyfp* was cultivated in different media to yield a range of growth rates in a Tecan Infinite M1000 Pro microplate reader. Growth was measured at a wavelength of 600 nm and EYFP fluorescence was detected at 527 nm using an excitation wavelength of 513 nm and a bandwidth of 5 nm. IPTG and chloramphenicol were added using the built-in injector of the Tecan Infinite M1000 Pro unit. Translation was stopped at 30 s intervals. Calculation of the translation elongation rate was done by fitting a linear regression to the square root plots[77]. The time for translation initiation was assumed to be 10 s as previously determined for *E. coli*[21].

**Self-replicator model and model calibration**
The mechanistic coarse-grained modelling approach used in this study follows the work of Bosdriesz et al.[50]. The self-replicator model provides insight into the connection between (intracellular) nutritional states and bacterial growth: the model considers associated governing rate parameters linked to protein synthesis (translation) and to growth-dependent transcriptional regulation of ribosome synthesis by means of a ppGpp-based mechanism. Herein, ribosomes forming a complex with uncharged tRNA, i.e. inactive non-translating ribosomes, provide a measure of ribosome inefficiency. The ordinary differential equation model was implemented in Matlab (Mathworks, Natick, MA, USA) and is provided in the Supplementary Model files.

Organism-specific model parameters of key interest in this study are the maximum translation rate $k_{R,max}$ and the ribosome production rate $\upsilon_{rrn}$ (referred to as the *rrn* operon transcription initiation rate in ref. 50). After calibrating the qualitative nutrient quality ($k_n$) for each of the studied steady-state growth conditions for *C. glutamicum*, rate parameters $k_{R,max}$ and $\upsilon_{rrn}$ were estimated from μ and $\Phi_R$ (fraction of ribosomal proteins) measurements, while the remaining model parameters were set to literature values (Supplementary Note 5, Supplementary Dataset 6). For model calibration, 10 data points were available (Supplementary Dataset 2). Each data point consists of the fraction of ribosomal protein ($\Phi_R$), derived from the fraction of total RNA per total protein, and the specific steady-state growth rate (μ), ranging from 0.07 to 0.94 h$^{-1}$, at which $\Phi_R$ was measured. Standard deviations for the measurements were determined for each data point for model-based error propagation. Analogously, the model

calibration procedure was performed for *E. coli* grown at 30 °C and for *E. coli* grown at 37 °C, based on publicly available data (see Supplementary Note 5C for details) and for additional experimental data for *E. coli* grown at 30 °C obtained in this work (Supplementary Dataset 4).

For parameter estimation, nonlinear weighted least-squares regression was applied. In short, the weighted residual sum-squared error between the model-predicted and measured data were minimized based on a global gradient-free optimization heuristic. Differential equation systems were solved with Matlab's ode15s solver. Standard deviations of the model parameters were calculated by parametric bootstrapping (500 runs with random initial parameter values). The $\chi^2$-test was applied to assess fit quality. All calculations were performed with custom Matlab scripts. Further details on the model, data processing, and calibration procedures are provided in the Supplementary Note 5. The resulting parameter settings of the three calibrated models used in this work are listed in Supplementary Datasets 6 and 7.

To investigate the response of intracellular ribosome production to an upshift of the growth rate in *C. glutamicum* in silico, the medium was switched from comparably poor media (CGXII+Glutamate, CGXII+EtOH, and CGXII + PYR) to a comparably rich medium (CGXII + GLU). To this end, steady-state growth rates μ and ribosomal protein fractions $\Phi_R$ were simulated with the calibrated *C. glutamicum* model for each of the three poor media, mimicked via appropriate $k_n$ values (Supplementary Dataset 10). The nutrient upshift was simulated as follows: first, nutrient-poor conditions were simulated. The resulting steady-state values were then used as initial conditions for simulating the nutrient-rich condition. Analogously, an upshift was simulated in *E. coli* at 30 °C from comparatively poor medium (MOPS + GLY) to a richer medium (MOPS + GLU). Results of the simulated pre-shift and post-shift growth rates of *C. glutamicum* and *E. coli* (at 30 °C) μ$_0$ and μ$_i$, respectively, were compared to results reported in literature for *E. coli* (at 30 °C)[6]. For maximal growth rate prediction ($\hat{\mu}_{max}$), the upper limit of the maximum specific growth rates for *C. glutamicum* and *E. coli* were predicted with the calibrated models by assuming a hypothetical "super-medium" nutritional quality value ($k_n$), being two orders of magnitude larger than the $k_n$ used for model calibration (Supplementary Dataset 8), while using the previously estimated $k_{R,max}$ and $\upsilon_{rrn}$ values.

**Reporting summary**
Further information on research design is available in the Nature Portfolio Reporting Summary linked to this article.

## Data availability
The data generated in this study are provided in the Supplementary Datasets 1–10, Source data files and in the Source Code (Supplementary Model Files and Supplementary Data File – Upshift Simulations). Any additional data can be obtained from the corresponding authors. Source data are provided with this paper.

## Code availability
The custom software *SurEmCo* can be found at https://github.com/modsim/suremco (https://doi.org/10.5281/zenodo.7991346). The modelling files can be found in the Source Code.

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

## Acknowledgements

The authors gratefully acknowledge the financial support of this study by the German Federal Ministry of Education and Research (BMBF; Grants 031A302C and 031B0918A) and the CLIB-Competence Center Biotechnology (CKB) funded by the European Regional Development Fund ERDF (grant number 34.EFRE-0300097). The authors thank Wolfgang Wiechert (Forschungszentrum Jülich) for fruitful discussions concerning the coarse-grained modelling effort as well as Julia Frunzke (Forschungszentrum Jülich) for providing the pEKEx2-*eyfp* plasmid.

## Author contributions

S.M. and M.B. designed the study. S.M., T.G., I.A. and J.H. performed the microscopy imaging acquisition. C.C.S. and K.N. developed the super-resolution emitter counter tool, SurEmCo. S.M., M.G. and R.T. contributed to the chemostat cultivations. S.M. and L.H. performed total RNA and protein extractions. S.M., N.T., J.T. and S.N. contributed to the translation elongation rate measurements. M.C. and K.N. adapted the self-replicator model and performed the simulations. S.M. performed the nutrient up-shit experiments. S.M., T.G., M.C., K.N. and M.B. analysed the data and wrote the manuscript as well as the Supplementary Information.

## Funding

## Competing interests

The authors declare no competing interests.
