## [Peer Review File · Nature Communications]

Growth-rate dependency of ribosome abundance and translation elongation rate in *Corynebacterium glutamicum* differs from that in *Escherichia coli*Reviewer #1 (Remarks to the Author):

In this paper, Matamouros et al. report ribosome localization, ribosome abundance, and translation elongation rate in *Corynebacterium glutamicum* over a range of growth rates. So far, relationships between these parameters have been measured in different growth conditions mostly in *Escherichia coli* and have been used to establish growth models for *E. coli*. In this paper, ribosome abundance was quantified in *C. glutamicum* by single-molecule localization microscopy, and its relationship with the growth rate followed a good agreement with that of R/P ratio measured by another assay. The authors found a critical difference between *C. glutamicum* and *E. coli*: *C. glutamicum* maintains relatively high ribosome copies even at a slow growth rate, but the rate of translation elongation decreases significantly as the growth rate decreases. A model-based approach could recapitulate these results based on the depletion of translation precursors. Nutrient up-shift experiments showed that the excess ribosomes during poor growth conditions allow the cells to quickly restart growth when conditions improve.

The difference to the well-established *E. coli* is interesting and is potentially of high impact. However, the key experiments need controls (see below Major question #1-2). Also, mechanistic explanations for how *C. glutamicum* and *E. coli* exhibit different patterns in ribosome abundance and translation elongation rate at slow growth are not addressed, limiting fundamental insights out of this paper.

Another major weakness of this paper is readability. The authors assumed that the readers know a lot about the subject matter, and it was difficult to understand some terms. For example, what are "active" ribosomes and "inactivated" ribosomes? Defining them at the molecular level (for example, is active ribosome the ribosome engaged in translation?) would help readers understand the authors' logic and conclusions. Also, authors should better explain their assumptions for different scenarios as well as expected outcomes (see my minor comment #2). Lastly, some of the non-trivial assays were presented too briefly in the main text. For example, "translation elongation rates and derived active ribosome fractions were measured for different growth rate using a fluorescence-based assay". It would be nice to mention what that assay is in 1-2 sentences in the main text. The same thing goes for the modeling section. Critical assumptions and input/output of the model should be clarified in the main text.

Major questions/concerns

- 1. Ribosome abundance was quantified from photo-activation of PAMCherry from fixed cells. Fixation is known to quench fluorescence and photo-activation cannot be 100%. Molecules outside the focal plane won't be imaged. These factors would contribute to the under-counting of PAMCherry. It would be nice to show a positive control case for which the copy number can be measured by another assay.**
- 2. Assay for translation elongation: Authors measured the time delay for the YFP fluorescence appearance after IPTG addition and considered a fixed initial delay in the translation initiation step (10 s) to extract translation elongation time. Authors should check if the initial delay (10 s) is actually constant under different growth conditions. The difference in speed can be explained by the difference in the initial delay. YFP gene is rather short, and a few seconds will make a big difference in speed.**
- 3. From nutrient up-shift experiments, authors claim that keeping excess ribosomes during poor nutrient conditions is useful to quickly restart growth when conditions improve. This argument should be true for any bacterial species. Why wouldn't *E. coli* adopt this strategy?**

Minor questions/comments

- 1. These lines need references:**
 - a. Line 52: "as growth rate increases, ribosomal mass fraction increases"**
 - b. Line 54: "ribosome synthesis is tightly controlled to avoid loss of fitness due to resource misallocation"**

- c. Line 138: Reference #29 should be included
- d. Line 195: Authors should cite 10.1128/jb.176.10.2807-2813.1994
- e. Line 203: "hibernation mechanisms"
- f. Line 288: Authors should cite 10.1126/science.1195691
- 2. These are sentences difficult to understand:
 - a. Line 62-64.
 - b. Line 142: Data for "the large variation" is missing.
 - c. Line 162: Please include a brief explanation for why R/P ratio can be a proxy for the ribosome abundance.
 - d. Line 179-181: A critical assumption is missing (I guess it is about translation precursors).
 - e. Line 187-188.
 - f. Line 210: What is the expected outcome of this assay in case there are defective ribosome particles?
 - g. Line 215: What is the implication of diauxic growth?
 - h. Line 226-229: It is difficult to understand the logic. If there were non-functional ribosome particles, what is expected? I don't see the two sentences in these lines are mutually exclusive.
 - i. Line 259: What is "it"?
 - j. Line 307-308: Another sentence should be added to explain this statement.
 - k. Line 362: Faster than what?
 - l. Line 368-370: Why do the famine-and-feast cycles result in growth rate optimization and not rapid growth recovery?
- 3. Figure 1: (b) What type of images are shown in the left column? Why are there are multiple puncta per cell in the DNA image?
- 4. Figure 2: (b-c) How many cells were used in the analysis?
- 5. Figure 3: What is your definition of stationary phase?
- 6. Two different strains (wt and MB001) were used in different assays, but whether different genotypes affect the results was not discussed.

Reviewer #2 (Remarks to the Author):

In this manuscript, the Authors analyze dependency of the ribosomal content in *C. glutamicum* cells on the growth rate. For this, they employ a single-molecule fluorescent reporter-based assay to detect ribosomes in single cells. Based on these single-cell results, as well as established methods to analyze ribosomal proteome fraction at population level, the Authors show that the proteome fraction, occupied by ribosomes in *C. glutamicum* is relatively stable at low growth, with a substantial increase when $\mu > 0.4/h$. In order to show that ribosomes at slow growth are functional, Authors performed nutritional up-shifts where they observed a fast response in terms of increased specific growth rate and ribosomal proteome fractions.

Major comments

1. The experimental design allows generating single-cell data at medium- or high-throughput. However, the authors barely use the opportunity to dissect the data more at the single-cell level: Fig 2b provides some insight into how many ribosomes are detected in single cells at different growth rates, but this is barely discussed in the text. I believe that a more in-depth analysis could strengthen the impact of this paper. For example, how do the actual distributions of ribosome count/cell look like for different growth rates? Figure 2 provides little quantitative insight due to overlap of data points. Additionally, in Figure 1, the authors also include the DNA channel. Has it been done for all the experiments? Would it be possible to correlate the ribosome content with the cell age in the exponentially-diving cells by relating the ribosome amount to the DNA content?

2. The central claim of the manuscript is that *C. glutamicum* exhibits different proteome allocation at low growth in terms of ribosomes than previously observed for *E. coli*. When analyzing the differences in elongation rates and active ribosome fraction (Fig 4), the authors compare the measurements of ribosomal proteome content of *E. coli* at 30/37°C (Fig 4a), with 30°C being the optimal temperature for *C. glutamicum*. The difference between *E. coli* and *C. glutamicum* in panel 4a at 30°C is not very different, and unfortunately, further analyses (panels 4b and 4c) do not have *E. coli* data in, or comparison is with the optimal *E. coli* temperature (37°C). Earlier in the manuscript, the Authors claim that “[an] upward deviation in the R/P ratio at lower growth <...> is reported for *E. coli* grown at 37°C” (L 177). Furthermore, authors suggest that at 30°C, elongation in *E. coli* happens at similar velocity (9 aa/s) as for *C. glutamicum*. Combining this information, one could claim that *C. glutamicum* ribosome allocation at 30°C does not drastically differ from the one in *E. coli* at 30°C, and the “more linear” trend we see in *E. coli* at 37°C is due to its adaptation to higher temperatures (to add, the authors themselves reflect on the Arrhenius’ law in the manuscript, see L 194). Eventually, I believe that the comparison that the authors performed might be not sufficient.

A potential scenario, if *E. coli* at 30°C behaved the same as *C. glutamicum* - suggesting temperature-dependent resource allocation - would be a very exciting insight in the field, indeed (although I do not really believe it). To resolve this, in an ideal case, a ribosome quantification experiment could be done for *E. coli* at 30°C. Alternatively, I suggest resorting to work on yeast (also grown at 30°C) where an inactive fraction was shown similar to *E. coli*. Also there the claim is made that an inactive fraction of ribosomes is used for rapid startup of growth when conditions improve (see e.g. Metzler-Raz, Elife). So how different is *C. glutamicum*, really? This is a crucial point, as it is the key message of the paper! The real difference appears to be in maintaining an inactive fraction at constant ribosome saturation (*E. coli*, yeast) versus low saturation of all ribosomes (*C. glutamicum*)? Please substantiate the claim of *C. glutamicum* being so different. Another approach that would improve the manuscript, is comparison of *E. coli* and *C. glutamicum* through the model, see next comment.

3. For *E. coli* we understand the ppGpp system maintains a constant saturation of ribosomes, ensuring a linear relationship between growth and ribosome fraction (work by Bosdriesz and Hwa). The fitting of the Bosdriesz model to the *C. glutamicum* data is interesting, and can describe the data, but how does it now explain the difference with *E. coli*? L 258 for example is rather cryptic. What are the differences in parameters between the organisms? What happens in the *E. coli* version of the model upon an upshift in glucose? I appreciate the effort of the authors to do theoretical analysis, and it seems to correspond well to the experimental data, but it now only shows that it can be fitted to the data (not really a surprise perhaps) and that it can produce rather trivial results from an upshift. What have you now really learned from the model, though? Comparison to *E. coli* would help I think.

Minor comment:

L49: I do not think our understanding of bacterial physiology depends on a few phenomenological growth laws. Just to mention all the chemostat work from Delft, DTU (or Julich), or more recently through genome-scale models by Palsson and co. Please weaken this statement.

L56-58: a linear relationship between ribosomes and growth rate implies a constant specific rate, not necessarily a maximal rate, nor does it indicate that ribosomes limit growth. This statement is indeed contradicted by the authors themselves at L63. Also L64, metabolite pools are never limiting, it is the synthesis rates of these pools. The authors are quite sloppy in this part of the introduction. Perhaps it is time to read up on the microbial physiology literature.

L 90: Authors constructed fusion proteins of some ribosomal subunits with fluorescent proteins. Even though Authors mention that ribosomes were still functional (comparable growth rate with WT), I wondered about the burden these fluorescent proteins introduce: how big are the fusion constructs, compared to the native ribosomal proteins bL19 and uS2? Where do these subunits appear in the ribosome structure? I miss

explicit reasoning why exactly these subunits were picked for manipulation.

L349: why would fast growth rate and the ability to survive starvation trade off?

Similarly, L 373, why would the "translational overcapacity" go at the cost of growth rate? This is not demonstrated and not obvious at low growth rates. Again, also yeast and E coli can recover quickly. I find the evolutionary and trade-off reasoning rather immature – as the physiological reason above. Please stop speculating and do a proper comparison.

RESPONSES TO REVIEWER COMMENTS

The authors would like to thank the reviewers for their very constructive comments! We have performed additional experiments, in particular for obtaining data for *E. coli* at 30°C, as requested by the reviewers, and revised the text according to their suggestions. Based on these data, the differences between *C. glutamicum* and *E. coli* at 30°C with respect to ribosome abundance, translation elongation rate, and active ribosome fraction could be confirmed.

Reviewer #1 (Remarks to the Author):

In this paper, Matamouros et al. report ribosome localization, ribosome abundance, and translation elongation rate in *Corynebacterium glutamicum* over a range of growth rates. So far, relationships between these parameters have been measured in different growth conditions mostly in *Escherichia coli* and have been used to establish growth models for *E. coli*. In this paper, ribosome abundance was quantified in *C. glutamicum* by single-molecule localization microscopy, and its relationship with the growth rate followed a good agreement with that of R/P ratio measured by another assay. The authors found a critical difference between *C. glutamicum* and *E. coli*: *C. glutamicum* maintains relatively high ribosome copies even at a slow growth rate, but the rate of translation elongation decreases significantly as the growth rate decreases. A model-based approach could recapitulate these results based on the depletion of translation precursors. Nutrient up-shift experiments showed that the excess ribosomes during poor growth conditions allow the cells to quickly restart growth when conditions improve.

The difference to the well-established *E. coli* is interesting and is potentially of high impact. However, the key experiments need controls (see below Major question #1-2). Also, mechanistic explanations for how *C. glutamicum* and *E. coli* exhibit different patterns in ribosome abundance and translation elongation rate at slow growth are not addressed, limiting fundamental insights out of this paper.

Another major weakness of this paper is readability. The authors assumed that the readers know a lot about the subject matter, and it was difficult to understand some terms. For example, what are “active” ribosomes and “inactivated” ribosomes? Defining them at the molecular level (for example, is active ribosome the ribosome engaged in translation?) would help readers understand the authors’ logic and conclusions. Also, authors should better explain their assumptions for different scenarios as well as expected outcomes (see my minor comment #2). Lastly, some of the non-trivial assays were presented too briefly in the main text. For example, “translation elongation rates and derived active ribosome fractions were measured for different growth rate using a fluorescence-based assay”. It would be nice to mention what that assay is in 1-2 sentences in the main text. The same thing goes for the modeling section. Critical assumptions and input/output of the model should be clarified in the main text.

Response:

Definitions, expected outcomes and additional information were added.

- Active ribosomes are defined as ribosomes engaged in peptide chain elongation (lines 52-54).
- Inactive ribosomes are defined as all ribosomes not participating in the translation process or unable to finish translation and yield the encoded protein (lines 196-198).

- The expected outcome for the different scenarios was added (lines 249-253).
- More information on the fluorescence-based translation elongation assay was added to the text (lines 202-205).
- A better description of the model including input and output were added (lines 279-287)

Major questions/concerns

1. Ribosome abundance was quantified from photo-activation of PAmCherry from fixed cells. Fixation is known to quench fluorescence and photo-activation cannot be 100%. Molecules outside the focal plane won't be imaged. These factors would contribute to the under-counting of PAmCherry. It would be nice to show a positive control case for which the copy number can be measured by another assay.

Response:

The reviewer is correct in pointing out experimental imperfections that would lead to an under-counting of molecules, e.g. incomplete photoactivation or molecules out of the focal plane of wide-field fluorescence microscope. We have taken great care to minimize these influences. For instance, we increased the photoactivation light intensity to very high values until no newly photoactivated PAmCherry molecules were detected, while, as described in the manuscript, we raised the photoactivation light power by very tiny amounts to keep the number of photoactivated molecules in the image low to prevent overlapping single molecule fluorescence patterns (and hence incomplete detection). This procedure allowed us – due to the high observation and photoactivation light powers used in combination with the low single molecule numbers per frame - also the photoactivation and detection of dimmer single molecules, where many of them will be placed near the ends of the focal region in z-direction, e.g. minimizing the effect of missing molecules outside of the focal layer. Nevertheless, the detection probability will not reach 100% as the reviewer stated.

There are, however, other short-comings of the experiment that would lead to an over-counting of PAmCherry molecules. All fluorescent proteins (FPs; in fact also the vast majority of organic fluorescent dyes) used in single molecule localization microscopy (SMLM) show blinking, i.e. periods of fluorescence interrupted by non-fluorescent periods. These so-called OFF-times can last anything between milliseconds to hours but were mostly found on the seconds time scale. While PAmCherry has been shown to be less affected by this phenomenon when compared to other FPs (see reference [1] at the end of this document), blinking occurs to a non-negligible fraction of PAmCherry molecules and with OFF-times in the magnitude of many seconds. Another reason for over-counting molecules originates from varying single molecule detection and identification efficiency by the SMLM fitting program due to varying noise contributions in individual frames. Typically, a PAmCherry molecule in our setup will be detected in a streak of consecutive 2 to ~15 frames and by post-analysis (described in Supplementary Note 1) is counted as one molecule. If a particular PAmCherry molecule has a low signal-to-noise and is near the identification limit, due to the stochastic nature of the emission process and temporal fluctuations of the different noise sources, such a molecule might not be identified in one or a few individual frames leading to an artificial double or multi-count. A similar effect can be caused by instabilities of the x- and y-position. If those would exceed the diameter of the single molecule image (ca. 30 nm) within an ON-period, the molecule would be counted as two

molecules. Again, we have tried to correct for this experimental imperfection by checking for single OFF-period frames and correct for them (reduce the count to 1).

Since it is not possible to correct for all these imperfections we rather did estimate – as suggested by the reviewer – the ribosome number per cell in an independent experiment for two of the conditions tested. If one compares the ribosome numbers estimated by SMLM and total RNA/cfu measurements one sees a fairly good agreement of the ribosome numbers detected by the two different and independent methods. We conclude from this agreement that the various experimental imperfections leading to under- and over-counting of PAmCherry molecules counterbalance/compensate each other and the numbers estimated by SMLM are correct. We apologize that we did not mention this comparison more clearly in the original text and therefore have added it as a new paragraph in the main text (lines 158-167): “To further check the accuracy of our SMLM-counting data, we estimated the number of ribosomes per cell from total RNA measurements and the mean colony forming units (cfu) per OD₆₀₀ for two of the growth conditions previously used. In exponentially growing cells the majority (~86%) of the total RNA in the cell is rRNA^{7,12,17}. The theoretical number of ribosomes can thus be calculated from the total RNA quantification from a known number of cells (calculated by determining the number of cfu OD₆₀₀⁻¹ mL⁻¹) and the rRNA mass of a single ribosome (2.53x10⁻¹⁵ mg; Supplementary Note 2). For cells growing exponentially in BHI+GLU medium and CGXII+GLU medium, 43,821 and 22,646 ribosomes/cell were calculated with this method, which is in very good agreement with the numbers determined by SMLM (45,207 and 25,770 ribosomes/cell).”

2. Assay for translation elongation: Authors measured the time delay for the YFP fluorescence appearance after IPTG addition and considered a fixed initial delay in the translation initiation step (10 s) to extract translation elongation time. Authors should check if the initial delay (10 s) is actually constant under different growth conditions. The difference in speed can be explained by the difference in the initial delay. YFP gene is rather short, and a few seconds will make a big difference in speed.

Response:

This is indeed a very interesting question and we agree with the reviewer in that EYFP (238 aa) is rather short for this assay. To address this concern we constructed a larger reporter where the mCherry fluorescent protein was translationally fused to the EYFP protein via a short linker resulting in a 508 aa protein (mCherry-linker-EYFP). We then performed the assay in the same manner as before, that is, by following the EYFP fluorescence and determining the time at which the first fluorescent molecules appear (lines 205-209). Following the same rationale as described in Zhu M. *et al* (see reference [2] at the end of this document), we determined the initiation time for four of the conditions we initially tested. The results showed that the time for translation initiation is around 11 s for *C. glutamicum* (which is close to the 10 s for *E. coli*) except for the slowest growth rate, where it reaches almost 14 s (lines 213-224 and Supplementary Table 3). The translation elongation rates were corrected using the newly determined times for translation initiation and comparison of the short construct (EYFP - 238 aa) with the long construct

(mCherry-L-EYFP – 508 aa) resulted in very similar translation elongation rates (In 209- 213, Supplementary Table 3).

3. From nutrient up-shift experiments, authors claim that keeping excess ribosomes during poor nutrient conditions is useful to quickly restart growth when conditions improve. This argument should be true for any bacterial species. Why wouldn't *E. coli* adopt this strategy?

Response:

Both *E. coli* and *C. glutamicum* keep an excess of ribosomes during slow growth. The difference lies on the fraction of ribosomes kept in an active state. *E. coli* adopts hibernation mechanisms to maintain a large portion of its ribosomes in an inactive state, releasing resources for the remaining ribosomes to keep working at a fast rate. In *C. glutamicum* we observe that the majority of ribosomes are still active during slow growth, which results in lower translation elongation rates (when compared to *E. coli*), probably due to scarcity of translation precursors. It is not clear what the advantage of one strategy over the other is.

Minor questions/comments

1. These lines need references:

a. Line 52: “as growth rate increases, ribosomal mass fraction increases”

Response:

References 8 and 12 were added (line 48):

- 8 Schaechter, M., Maaløe, O. & Kjeldgaard, N. O. Dependency on medium and temperature of cell size and chemical composition during balanced growth of *Salmonella typhimurium*. *J Gen Microbiol* **19**, 592-606 (1958).
- 12 Neidhardt, F. C. & Magasanik, B. Studies on the role of ribonucleic acid in the growth of bacteria. *Biochim Biophys Acta* **42**, 99-116 (1960).

b. Line 54: “ribosome synthesis is tightly controlled to avoid loss of fitness due to resource misallocation”

Response:

References 13-15 (line 50):

- 13 Wilson, D. N. & Nierhaus, K. H. The weird and wonderful world of bacterial ribosome regulation. *Crit Rev Biochem Mol* **42**, 187-219 (2007).
- 14 Scott, M., Klumpp, S., Mateescu, E. M. & Hwa, T. Emergence of robust growth laws from optimal regulation of ribosome synthesis. *Mol Syst Biol* **10**, 747 (2014).
- 15 Stevenson, B. S. & Schmidt, T. M. Growth rate-dependent accumulation of RNA from plasmid-borne rRNA operons in *Escherichia coli*. *J Bacteriol* **180**, 1970-1972 (1998).

c. Line 138: Reference #29 should be included

Reponse:

The reference was included (line 136):

33 Gray, W. T. *et al.* Nucleoid size scaling and intracellular organization of translation across bacteria. *Cell* **177**, 1632-1648 e1620 (2019).

d. Line 195: Authors should cite 10.1128/jb.176.10.2807-2813.1994

Response:

The reference was included (line 233):

48 Vogel, U. & Jensen, K. F. The RNA chain elongation rate in *Escherichia coli* depends on the growth rate. *J Bacteriol* **176**, 2807-2813 (1994).

e. Line 203: “hibernation mechanisms”

Response:

Reference 31 was included (line 242):

21 Dai, X. *et al.* Reduction of translating ribosomes enables *Escherichia coli* to maintain elongation rates during slow growth. *Nat Microbiol* **2**, 16231 (2016).

f. Line 288: Authors should cite 10.1126/science.1195691

Response:

The reference was included (line 358):

54 Nevo-Dinur, K., Nussbaum-Shochat, A., Ben-Yehuda, S. & Amster-Choder, O. Translation-independent localization of mRNA in *E. coli*. *Science* **331**, 1081-1084 (2011).

2. These are sentences difficult to understand:

a. Line 62-64.

Response:

The statement “In *E. coli* the translation rate decreases by about 50% from fast to very low growth rates [11, 15-18] denoting that ribosomes are not always working in saturating substrate conditions.” was changed to (lines 58-63): “In *E. coli* the translation elongation rate (k) decreases by about 50% from fast ($k = 16-17 \text{ aa s}^{-1}$, $\mu > 1 \text{ h}^{-1}$) to very low growth rates ($k = 9 \text{ aa s}^{-1}$, $\mu = 0.035 \text{ h}^{-1}$)¹⁶⁻²¹, which shows that the translation elongation rate is not constant. The availability of aminoacyl-tRNAs as well as elongation factors and GTP may be reduced in certain conditions, such as during nutrient deprivation or by slow diffusion in the crowded cytoplasm²²⁻²⁴, leading to a decrease in the elongation rate.”

b. Line 142: Data for “the large variation” is missing.

Response:

We added panel b (box and whiskers plot) to Fig. 2 and a Supplemental Fig. 3 to better show the variation of the number of ribosomes per cell found for the different media (growth rates) tested.

c. Line 162: Please include a brief explanation for why R/P ratio can be a proxy for the ribosome abundance.

Response:

A short explanation has been included in lines 173-177: “The R/P ratio is a surrogate and widely used method for the determination of ribosome abundance, since in exponentially growing cells the protein content remains largely invariant with the growth rate and the number of ribosomes is proportional to the rRNA, which comprises the majority of the total RNA of the cell ^{7,10,12,39,40}.”

d. Line 179-181: A critical assumption is missing (I guess it is about translation precursors).

Response:

The sentence/paragraph “In *E. coli*, deviation of the R/P ratio from linearity during slow growth is associated with a strong reduction of the active ribosome pool to allow maintenance of high translation elongation rates [18]” was reformulated to (lines 191-199):

“A deviation from the linear relationship between R/P ratio and μ has also been reported for *E. coli* grown at 37 °C and growth rates $<0.7 \text{ h}^{-1}$ ²¹, but it is much weaker than the one observed for *C. glutamicum* (Fig. 4a). In *E. coli*, the deviation is associated with a strong reduction of the active ribosome pool, hypothesized by Dai *et al.* ²¹ to be caused by either hibernation mechanisms ⁴³, ppGpp-regulated translation initiation inhibition ⁴⁴, or by passive translation abortion caused by ribosome stalling due to a decreased availability of translation precursors ⁴⁵. All ribosomes not participating in the translation process or unable to finish translation and yield the encoded protein are then considered inactive ²¹. By reduction of the number of active ribosomes *E. coli* allows the preservation of high translation elongation rates during slow growth ²¹.”

e. Line 187-188.

Response:

Original sentence: “The lower translation rates at 30 °C for both organisms are well reflected in the steeper slopes of the linear Rb/ μ correlation (Fig. 4a) [13].”

The following sentence was added to clarify the statement (lines 229-232): “It means that in order to support a certain growth rate at 30 °C, cells require more ribosomes than at 37 °C due to

the lower translation elongation rate, which follows an Arrhenius kinetics for growth temperatures between 23-44 °C⁴⁷.”

f. Line 210: What is the expected outcome of this assay in case there are defective ribosome particles?

Response:

Original sentence: “To exclude that the high abundance of ribosomes as well as the marked decrease in the translation elongation rate observed at slow growth rates for *C. glutamicum* was due to the accumulation of defective or misassembled ribosome particles, nutrient upshift experiments were performed.”

The following sentence was added (lines 249-253): “In the presence of defective ribosomal particles, we expected to observe an initial phase after the upshift dominated by the synthesis of functional ribosomes and consequently a delay in the increase of the instantaneous growth rate. On the contrary, if the ribosomes present in the cell are functional, they should allow for an immediate increase in growth rate.”

g. Line 215: What is the implication of diauxic growth?

Response:

For this purpose, diauxic growth is not relevant and therefore that information was removed from the text.

h. Line 226-229: It is difficult to understand the logic. If there were non-functional ribosome particles, what is expected? I don't see the two sentences in these lines are mutually exclusive.

Response:

Original statement “These results show that in both conditions there is a pre-existing ribosome overcapacity that allows cells to instantly respond to the improved growth conditions. Therefore, we conclude that the high ribosome abundance observed for *C. glutamicum* during slow growth is not due to the accumulation of non-functional ribosomal particles.”

We agree that is indeed not clear, so we attempt to clarify what we mean. If the cells only possessed the ribosomes needed to support the current growth rate, in this case 0.06 h⁻¹ and 0.17 h⁻¹, they would need time to synthesize additional ribosomes to support the higher growth rates on the second carbon source added. On the contrary, if the cells already possess “extra” ribosomes we would expect the jump in growth rate to occur immediately.

We have simplified our statement as follows (lines 266-268): “These results show that in both conditions there is a pre-existing ribosome overcapacity that allows cells to instantly respond to the improved growth conditions shown by the immediate increase in the instantaneous growth rate to 0.3 h⁻¹.”

i. Line 259: What is “it”?

Response:

“it” has been substituted by “the SR model for *E. coli*” (lines 312-313).

j. Line 307-308: Another sentence should be added to explain this statement.

Response:

The sentence “At $\mu < 0.4 \text{ h}^{-1}$, faster growth of *C. glutamicum* is thus mainly achieved by the increase in the rate of protein synthesis rather than by an increase in ribosome abundance.” was modified to (lines 373-376):

“At μ between $0.07\text{-}0.29 \text{ h}^{-1}$, faster growth of *C. glutamicum* is thus mainly achieved by the observed progressive increase in the rate of translation elongation, from 1.81 to 6.96 aa s^{-1} , rather than by an increase in ribosome abundance, which remains constant for $\mu < 0.4 \text{ h}^{-1}$.”

k. Line 362: Faster than what?

Response:

This section of the discussion was removed.

l. Line 368-370: Why do the famine-and-feast cycles result in growth rate optimization and not rapid growth recovery?

Response:

Mori et al. (see reference [3] at the end of this document) studied the impact of an overcapacity of the translation machinery on the kinetics of growth rate recovery. He found that the ribosome overcapacity of *E. coli* is optimal for the time period when rich nutrients are present (feast) in the mammalian gut. In their simulations, strains with large overcapacity perform better right after the shift, but the cost associated with maintaining a large translational overcapacity reduces the maximum possible growth rate during the feast. In the case of *E. coli*, its overcapacity appears to be optimal for a feast period of a couple of hours, roughly the duration of the feast period in the gut. The authors suggest that the translational overcapacity for different organisms could be related to the environmental niche where they evolved, and one of the determinant factors could be the duration of the high nutrient period routinely encountered. Since we do not show any results that can support this idea we have removed this hypothesis from the discussion. Furthermore, as reviewer 2 suggested, a proper comparison should be performed to verify it.

3. Figure 1: (b) What type of images are shown in the left column?

Response:

The images are bright-field and wide-field fluorescence microscopy images. The legend for Fig. 1 was updated (lines 945-947) to: “The first pair of panels on the left are bright-field microscopy images. The three pairs of panels on the right are wide-field fluorescence microscopy images.”

Why are there are multiple puncta per cell in the DNA image?

Response:

The multiple puncta per cell in the DNA image correspond to the nucleic acid-rich regions or nucleoid that in rapidly growing cells can adopt an irregular shape. A similar nucleoid distribution can be observed for other bacterial species (see references [4-9] at the end of this document).

4. Figure 2: (b-c) How many cells were used in the analysis?

Response:

The number of cells analyzed was previously only given in Supplemental Table 1. To make this information more accessible, we have included it in the new Fig. 2b.

5. Figure 3: What is your definition of stationary phase?

Response:

We define stationary phase as a period where bacterial growth stops but cells are still metabolically active. In this case, samples were taken at 24 h where, for all media tested, the biomass had been constant for at least 8-10 h with no detectable cell death.

6. Two different strains (wt and MB001) were used in different assays, but whether different genotypes affect the results was not discussed.

Response:

MB001(DE3) is a genome-reduced strain derived from the wt (ATCC13032) with the DE3 region from *E. coli* BL21(DE3) integrated into its chromosome. Growth of these two strains is very similar. This information was added in lines 418-421:

“The strains used in this study were the *C. glutamicum* type strain ATCC 13032 (wt) or its derivatives MB001(DE3), a prophage-free strain, which allows for IPTG-inducible T7-RNA polymerase-dependent gene expression⁸⁹ that shows a very similar growth behaviour as the wt⁹⁰, and EVO5, a laboratory-evolved faster-growing strain⁵⁹.”

Reviewer #2 (Remarks to the Author):

In this manuscript, the Authors analyze dependency of the ribosomal content in *C. glutamicum* cells on the growth rate. For this, they employ a single-molecule fluorescent reporter-based assay to detect ribosomes in single cells. Based on these single-cell results, as well as established methods to analyze ribosomal proteome fraction at population level, the Authors show that the proteome fraction, occupied by ribosomes in *C. glutamicum* is relatively stable at low growth, with a substantial increase when $\mu > 0.4/h$. In order to show that ribosomes at slow growth are functional, Authors performed nutritional up-shifts where they observed a fast response in terms of increased specific growth rate and ribosomal proteome fractions.

Major comments

1. The experimental design allows generating single-cell data at medium- or high-throughput. However, the authors barely use the opportunity to dissect the data more at the single-cell level: Fig 2b provides some insight into how many ribosomes are detected in single cells at different growth rates, but this is barely discussed in the text. I believe that a more in-depth analysis could strengthen the impact of this paper. For example, how do the actual distributions of ribosome count/cell look like for different growth rates?

Figure 2 provides little quantitative insight due to overlap of data points.

Response:

We agree with the reviewer and have therefore included a new figure panel (Fig. 2b) with a box and whiskers plot where the distribution of the ribosome number per cell can be examined in more detail for each tested condition/growth rate. In addition, we have also added the individual data for ribosome numbers/cell for each growth condition analyzed in Supplemental Fig. 3. These data were used to create Fig. 2c.

Additionally, in Figure 1, the authors also include the DNA channel. Has it been done for all the experiments? Would it be possible to correlate the ribosome content with the cell age in the exponentially-diving cells by relating the ribosome amount to the DNA content?

Response:

When planning and preparing this extended SMLM experiment series, our plan was (as the reviewer requests) to include experiments that could allow us to determine DNA localization and quantification derived from super-resolution microscopy. For this purpose, several DNA stains with different excitation and emission characteristics (DAPI, Hoechst, different SYTOX dyes, ADDvanced, SiRDNA, and more) were tested in our SMLM setup. However, none was suitable for good SMLM measurements – be it for super-resolution imaging or counting. The best results in terms of detected fluorescence intensity versus autofluorescence background were obtained when using Sytox Orange. Since the fluorescence emission of photoconverted PAmCherry overlaps with that of Sytox Orange, we used a bacterial strain encoding a bL19-EYFP translational fusion (ribosome) to be able to co-visualize the distribution of ribosomes and cellular DNA for the same cell. Therefore, the two-color wide-field fluorescence images obtained in this way are only suitable for visualizing DNA and ribosome distribution and, unfortunately, do not allow quantification of DNA amount or ribosome numbers.

2. The central claim of the manuscript is that *C. glutamicum* exhibits different proteome allocation at low growth in terms of ribosomes than previously observed for *E. coli*. When analyzing the differences in elongation rates and active ribosome fraction (Fig 4), the authors compare the measurements of ribosomal proteome content of *E. coli* at 30/37°C (Fig 4a), with 30°C being the optimal temperature for *C. glutamicum*. The difference between *E. coli* and *C. glutamicum* in panel 4a at 30°C is not very different, and unfortunately, further analyses (panels 4b and 4c) do not have *E. coli* data in, or comparison is with the optimal *E. coli* temperature (37°C). Earlier in the manuscript, the Authors claim that “[an] upward deviation in the R/P ratio at lower growth <...> is reported for *E. coli* grown at 37°C” (L 177). Furthermore, authors suggest that at 30°C, elongation in *E. coli* happens at similar velocity (9 aa/s) as for *C. glutamicum*.

Combining this information, one could claim that *C. glutamicum* ribosome allocation at 30°C does not drastically differ from the one in *E. coli* at 30°C, and the “more linear” trend we see in *E. coli* at 37°C is due to its adaptation to higher temperatures (to add, the authors themselves reflect on the Arrhenius’ law in the manuscript, see L 194). Eventually, I believe that the comparison that the authors performed might be not sufficient.

A potential scenario, if *E. coli* at 30°C behaved the same as *C. glutamicum* - suggesting temperature-dependent resource allocation - would be a very exciting insight in the field, indeed (although I do not really believe it). To resolve this, in an ideal case, a ribosome quantification experiment could be done for *E. coli* at 30°C. Alternatively, I suggest resorting to work on yeast (also grown at 30°C) where an inactive fraction was shown similar to *E. coli*. Also there the claim is made that an inactive fraction of ribosomes is used for rapid startup of growth when conditions improve (see e.g. Metzl-Raz, Elife). So how different is *C. glutamicum*, really? This is a crucial point, as it is the key message of the paper! The real difference appears to be in maintaining an inactive fraction at constant ribosome saturation (*E. coli*, yeast) versus low saturation of all ribosomes (*C. glutamicum*)? Please substantiate the claim of *C. glutamicum* being so different.

Response:

As suggested by the reviewer we determined R/P ratio as well as translation elongation rates for *E. coli* at 30 °C for different growth rates, which allowed us to determine the fraction of active ribosomes for *E. coli* 30 °C (Supplemental Table 4). As shown in Fig. 4a, R/P ratio for *C. glutamicum* and *E. coli* is very similar at higher growth rates, but at lower growth rates (< 0.4 h⁻¹), a clear deviation from linearity is only seen for *C. glutamicum*, but not for *E. coli*. Also, the translation elongation rate for *E. coli* at 30 °C was always around 9 aa s⁻¹, while for *C. glutamicum* it decreases 5-fold from the faster to the slowest growth rate at which it was measured (Fig. 4b). Finally, the active ribosome fraction calculated for *E. coli* 30 °C is close to the one determined by Dai *et al.* for *E. coli* at 37 °C. For growth rates below 0.2 h⁻¹ we observe a clear difference between the active ribosome fraction present in *E. coli* and *C. glutamicum* (Fig. 4c).

3. Another approach that would improve the manuscript, is comparison of *E. coli* and *C. glutamicum* through the model. For *E. coli* we understand the ppGpp system maintains a constant saturation of ribosomes, ensuring a linear relationship between growth and ribosome fraction (work by Bosdriesz and Hwa). The fitting of the Bosdriesz model to the *C. glutamicum* data is interesting, and can describe the data, but how does it now explain the difference with *E. coli*? L

258 for example is rather cryptic. What are the differences in parameters between the organisms? What happens in the *E. coli* version of the model upon an upshift in glucose? I appreciate the effort of the authors to do theoretical analysis, and it seems to correspond well to the experimental data, but it now only shows that it can be fitted to the data (not really a surprise perhaps) and that it can produce rather trivial results from an upshift. What have you now really learned from the model, though? Comparison to *E. coli* would help I think.

Response:

We appreciate the comment and agree that the description of our reasoning can be improved. The self-replicator (SR) model we are using was formulated for *E. coli* at moderate to high growth regimes ($\mu > 0.5 \text{ h}^{-1}$), where essentially all ribosomes are actively translating. For *E. coli* it is known, however, that at lower growth rates a “ribosomal hibernation mechanism” kicks in, which inactivates a proportion of the ribosomes, while keeping the remaining ribosomes translating at relatively high rate (see reference [10] at the end of this document). Consequently, the SR model calibrated with *E. coli* data is not able to explain the data for growth rates below 0.3 h^{-1} ($30 \text{ }^\circ\text{C}$) and 0.5 h^{-1} ($37 \text{ }^\circ\text{C}$), respectively (Supplementary Note 6). In contrast, after recalibration (ribosome fractions), the SR-*C. glutamicum* model explained the whole range of observed growth rates between 0.07 and 0.94 h^{-1} , suggesting that in these regimes no additional adaptation mechanism is activated. Since data for slow growth rates below 0.07 h^{-1} are unavailable, we cannot exclude that such ribosomal hibernation mechanism is present in *C. glutamicum* in this extreme regime. Summarizing, we concluded that SR modelling suggests that *C. glutamicum* maintains a large proportion of ribosomes active over a broad range of growth rates, which is far larger than that of *E. coli*.

Turning to the model parameters, parameter estimation revealed differences in the maximum *rrn* initiation rate between the two organisms, which turns out to be 3-fold the rate in *E. coli*. Here, results show that ribosomal resource allocation is not temperature-dependent, *i.e.*, *E. coli* applies ribosome deactivation at low growth rates both at 30 and $37 \text{ }^\circ\text{C}$.

In silico upshift experiments were performed with the models, calibrated with *E. coli* and *C. glutamicum* data, respectively, in growth regimes, where all ribosomes are actively translating. Simulation results are qualitatively and quantitatively similar when upshifts were simulated to the same post-shift medium (*i.e.* same nutrient quality k_n after the shift). Under these conditions, we do not see a difference between *E. coli* and *C. glutamicum*.

Minor comment:

L49: I do not think our understanding of bacterial physiology depends on a few phenomenological growth laws. Just to mention all the chemostat work from Delft, DTU (or Julich), or more recently through genome-scale models by Palsson and co. Please weaken this statement.

Response:

We agree with the reviewer and have changed the statement from : “ Our understanding of bacterial physiology relies on a few phenomenological growth laws...” to (lines 45-46):

“Our understanding of bacterial physiology includes a number phenomenological growth laws...”

L56-58: a linear relationship between ribosomes and growth rate implies a constant specific rate, not necessarily a maximal rate, nor does it indicate that ribosomes limit growth. This statement is indeed contradicted by the authors themselves at L63. Also L64, metabolite pools are never limiting, it is the synthesis rates of these pools. The authors are quite sloppy in this part of the introduction. Perhaps it is time to read up on the microbial physiology literature.

Response:

The reviewer is right and we have revised the text accordingly (lines 50-63):

“A linear correlation between ribosome abundance and growth rate (Rb/μ) has been reported for several organisms, which highlights the importance of protein synthesis to cellular growth¹⁰. This proportionality relies on ribosomes translating at a constant rate⁸. Growth rate therefore depends on the number of active ribosomes (*i.e.*, ribosomes engaged in peptide chain elongation) present in the cell. However, it is known that this linear correlation is only observed for moderate to fast growth rates ($\mu > 0.35 \text{ h}^{-1}$)⁴, as a continuous linear decrease in the number of ribosomes as growth slows down would leave the cells with too few ribosomes to restart growth⁵. Most comprehensive studies on this subject have been performed on the fast-growing model organism *Escherichia coli*. In *E. coli* the translation elongation rate (k) decreases by about 50% from fast ($k = 16\text{-}17 \text{ aa s}^{-1}$, $\mu > 1 \text{ h}^{-1}$) to very low growth rates ($k = 9 \text{ aa s}^{-1}$, $\mu = 0.035 \text{ h}^{-1}$)¹⁶⁻²¹, which shows that the translation elongation rate is not constant. The availability of aminoacyl-tRNAs as well as elongation factors and GTP may be reduced in certain conditions, such as during nutrient deprivation or by slow diffusion in the crowded cytoplasm²²⁻²⁴, leading to a decrease in the elongation rate.”

L 90: Authors constructed fusion proteins of some ribosomal subunits with fluorescent proteins. Even though Authors mention that ribosomes were still functional (comparable growth rate with WT), I wondered about the burden these fluorescent proteins introduce: how big are the fusion constructs, compared to the native ribosomal proteins bL19 and uS2? Where do these subunits appear in the ribosome structure? I miss explicit reasoning why exactly these subunits were picked for manipulation.

Response:

The reviewer is correct and relevant information was missing. bL19 and uS2 proteins are 13 kDa and 29 kDa, respectively, whereas EYFP and PA-mCherry are both ~27 kDa. As shown in Fig. 1, growth of the labeled strain compared to that of the wt does not appear to be affected.

In order to explain why we choose bL19 and uS2, we added the following text (lines 80-83):

“These ribosomal proteins, which are located at the ribosome surface and incorporated at a late stage of ribosome assembly²⁵, were chosen because they have been successfully tagged with fluorescent proteins in *E. coli* and the resulting strains showed normal growth and no ribosome assembly defects^{26,27}.”

L349: why would fast growth rate and the ability to survive starvation trade off? Similarly, L 373, why would the “translational overcapacity” go at the cost of growth rate? This is not demonstrated and not obvious at low growth rates. Again, also yeast and E coli can recover quickly. I find the evolutionary and trade-off reasoning rather immature – as the physiological reason above. Please stop speculating and do a proper comparison.

Response:

We agree with the reviewer that this portion of the discussion was not clear and too speculative. We would like, however, try to clarify the different points.

In the literature there are a few examples that illustrate this trade-off between adaptation to starvation and decreased growth rate. For example, it has been shown that *E. coli* cultures adapted to prolonged starvation generated cells that are less “fit” than the parental strains when grown in fresh medium in a competition assay (see reference [11] at the end of this document). In a different study, *E. coli* adapted to periods of histidine starvation was shown to be able to grow better in medium containing lower histidine concentration but unable to reach the growth rate of the parental strain in nutrient-rich medium. This phenotype was attributed to an enhanced affinity for low amounts of histidine as well as alterations in the ribosome that resulted in lower translation elongation rates possibly to avoid fast depletion of this amino acid (see reference [12] at the end of this document).

In order to prepare for periods of starvation cells need to have reserves in the form of energy, metabolic enzymes, translation machinery, and others. The synthesis of these reserves can allow the cells to not only survive periods of starvation but also quickly respond to changes in the environment. However, the synthesis of these reserves comes at the expense of a higher theoretical maximal growth rate during steady state growth if the cells were to use all of their resources to synthesize only what they needed for growth in that particular environment. The translational overcapacity in particular, allows the cells to respond quickly to a change in environment, however the protein cost associated with it limits the exponential growth rate of an organism since it is producing more ribosomes and other translation associated factors than what it would need for growth in that particular condition. We did not intend to imply that *E. coli* does not recover quickly. What Mori *et al.* showed is that the overcapacity in *E. coli* appears to be optimal for the duration of the periods of high nutrient abundance encountered by *E. coli*. We then speculated if the high ribosome abundance for *C. glutamicum* that we observe during slow growth is an evolutionary adaptation to the environmental niche where it evolved. If, we consider Mori *et al.*'s prediction the organisms with high translational overcapacity should in principle show faster recovery than those with lower translational overcapacity at the expense of lower growth rate during steady state growth.

We agree that a proper comparison should be performed, especially at very low growth rates and therefore have removed this portion of the discussion.

References

1. Durisic, N., et al., *Single-molecule evaluation of fluorescent protein photoactivation efficiency using an in vivo nanotemplate*. Nat Methods, 2014. **11**(2): p. 156-62.
2. Zhu, M., X. Dai, and Y.P. Wang, *Real time determination of bacterial in vivo ribosome translation elongation speed based on LacZalpha complementation system*. Nucleic Acids Res, 2016. **44**(20): p. e155.
3. Mori, M., et al., *Quantifying the benefit of a proteome reserve in fluctuating environments*. Nat Commun, 2017. **8**(1): p. 1225.
4. Bakshi, S., et al., *Superresolution imaging of ribosomes and RNA polymerase in live Escherichia coli cells*. Mol Microbiol, 2012. **85**(1): p. 21-38.
5. Chai, Q., et al., *Organization of ribosomes and nucleoids in Escherichia coli cells during growth and in quiescence*. J Biol Chem, 2014. **289**(16): p. 11342-52.
6. Gray, W.T., et al., *Nucleoid size scaling and intracellular organization of translation across bacteria*. Cell, 2019. **177**(6): p. 1632-1648 e20.
7. Kim, J., et al., *Spatial organization of the gene expression hardware in Pseudomonas putida*. Environ Microbiol, 2019. **21**(5): p. 1645-1658.
8. Lewis, P.J., S.D. Thaker, and J. Errington, *Compartmentalization of transcription and translation in Bacillus subtilis*. EMBO J, 2000. **19**(4): p. 710-8.
9. Sanamrad, A., et al., *Single-particle tracking reveals that free ribosomal subunits are not excluded from the Escherichia coli nucleoid*. Proc Natl Acad Sci U S A, 2014. **111**(31): p. 11413-8.
10. Dai, X., et al., *Reduction of translating ribosomes enables Escherichia coli to maintain elongation rates during slow growth*. Nat Microbiol, 2016. **2**: p. 16231.
11. Vasi, F.K. and R.E. Lenski, *Ecological strategies and fitness tradeoffs in Escherichia coli mutants adapted to prolonged starvation*. Journal of Genetics, 1999. **78**(1): p. 43-49.
12. Ying, B.W., et al., *Evolutionary Consequence of a Trade-Off between Growth and Maintenance along with Ribosomal Damages*. Plos One, 2015. **10**(8).

Reviewer #1 (Remarks to the Author):

In this revised paper, Matamouros et al. performed control experiments to support their original results. They also made the manuscript much easier to read. Since this is a review for revision, I will just leave two remaining concerns.

1. Ribosome abundance was quantified from photo-activation of PAmCherry from fixed cells. As they mentioned in the rebuttal, there are technical issues that contribute to over- and under-counting of fluorescent molecules. In addition to what they listed, chemical fixation results in undercounting. PAmcherry is known to have internal start codon, producing free molecules (that are not fused to ribosomes in this case). Since cells are fixed, these molecules would be counted as ribosomes and result in overcounting. I am concerned that this study, if published this way, will be used by other researchers to support the use of PA-mCherry imaging as an accurate method for counting proteins. Therefore, I suggest authors tone it down and use the PA-mCherry results as approximated ribosome numbers.

That being said, I do not think PA-mCherry results affect their conclusion because the R/P ratio measurement supports their conclusion.

2. It was nice to see that the authors made a plasmid with mCherry and eYFP in tandem to measure translation elongation rate. Using this construct, they observed that translation elongation can be incredibly slow. However, such a result is possible (especially when precursors are depleted, as the authors proposed) if the ribosome does not translate to the eYFP region. In other words, it is possible that the rate of translation elongation is not very different in slow growth conditions in *C. glutamicum*.

The rates of translation elongation in *C. glutamicum* are shown to be different from those of *E. coli*, but that could be because they used mCherry-eYFP tandem construct for *C. glutamicum* and a single eYFP construct for *E. coli*.

Since active ribosome fraction depends on the rate of translation elongation, it is important to calculate translation elongation rate accurately.

3. Still some places need clarification:

Line228-229: It is difficult to understand why the lower translation rates are "well-reflected" in the steeper slopes of the linear part of the Rb/u correlation.

Line235-236: Authors wrote as if there is a causal relationship between reduced fraction of active ribosome (cause) and high translation rate (result). However, I am not sure about the causality.

Reviewer #2 (Remarks to the Author):

I thank the authors for their extensive revision and answers to the questions I had. I am happy with the improvements.

REVIEWERS' COMMENTS

Reviewer #1 (Remarks to the Author):

In this revised paper, Matamouros et al. performed control experiments to support their original results. They also made the manuscript much easier to read. Since this is a review for revision, I will just leave two remaining concerns.

1. Ribosome abundance was quantified from photo-activation of PAmCherry from fixed cells. As they mentioned in the rebuttal, there are technical issues that contribute to over- and under-counting of fluorescent molecules. In addition to what they listed, chemical fixation results in undercounting. PAmcherry is known to have internal start codon, producing free molecules (that are not fused to ribosomes in this case). Since cells are fixed, these molecules would be counted as ribosomes and result in overcounting. I am concerned that this study, if published this way, will be used by other researchers to support the use of PA-mCherry imaging as an accurate method for counting proteins. Therefore, I suggest authors tone it down and use the PA-mCherry results as approximated ribosome numbers. That being said, I do not think PA-mCherry results affect their conclusion because the R/P ratio measurement supports their conclusion.

Free PAmCherry was not detected by western blot using an antibody directed against mCherry in the crude extracts of cells carrying the bS2-PAmCherry protein fusion. However, the reviewer is right and we cannot exclude the possibility that free PAmCherry molecules can be formed at a level that cannot be detected by western blot and in this way contribute to the overestimation of the ribosome numbers. We have, therefore, followed the reviewer's suggestion and used the term "approximate ribosome numbers" where appropriate.

2. It was nice to see that the authors made a plasmid with mCherry and eYFP in tandem to measure translation elongation rate. Using this construct, they observed that translation elongation can be incredibly slow. However, such a result is possible (especially when precursors are depleted, as the authors proposed) if the ribosome does not translate to the eYFP region. In other words, it is possible that the rate of translation elongation is not very different in slow growth conditions in *C. glutamicum*. The rates of translation elongation in *C. glutamicum* are shown to be different from those of *E. coli*, but that could be because they used mCherry-eYFP tandem construct for *C. glutamicum* and a single eYFP construct for *E. coli*.

Since active ribosome fraction depends on the rate of translation elongation, it is important to calculate translation elongation rate accurately.

The single EYFP reporter construct was used to measure translation elongation rate for both *E. coli* and *C. glutamicum*. The mCherry-EYFP tandem construct was only used to verify the results obtained with the single EYFP reporter and to allow for calculation of the time for translation initiation.

3. Still some places need clarification:

Line228-229: It is difficult to understand why the lower translation rates are “well-reflected” in the steeper slopes of the linear part of the Rb/ μ correlation.

To try to improve clarity the statement was modified to: “ In order to support a certain growth rate at 30 °C, cells require more ribosomes than at 37 °C due to the lower translation elongation rate, which follows an Arrhenius kinetics for growth temperatures between 23-44 °C⁴⁷. This is well reflected in the steeper slopes of the linear part of the Rb/ μ correlation (Fig. 4a)¹⁰. Such steeper slopes are also observed in slow-translation mutants when compared to the parental strain¹⁰.”

Line235-236: Authors wrote as if there is a causal relationship between reduced fraction of active ribosome (cause) and high translation rate (result). However, I am not sure about the causality.

We had based our statement in the study of Dai, X. et al. Reduction of translating ribosomes enables *Escherichia coli* to maintain elongation rates during slow growth. *Nat Microbiol* 2, 16231 (2016). However, we understand the reviewer’s concerns and have rephrased it to “In *E. coli* it is observed that during slow growth relatively high translation rates are maintained while the fraction of active ribosomes is reduced to under 20%²¹ (Fig. 4c).”

Reviewer #2 (Remarks to the Author):

I thank the authors for their extensive revision and answers to the questions I had. I am happy with the improvements.